# Germline T cell receptor exchange results in physiological T cell development and function

Meagan R. Rollins [1,2], Jackson F. Raynor[1,2], Ebony A. Miller[1,2,9], Jonah Z. Butler[1,2], Ellen J. Spartz[1,2,8], Walker S. Lahr[3,4,5,6], Yun You[7], Adam L. Burrack[1,2], Branden S. Moriarity[3,4,5,6], Beau R. Webber [3,4,5,6,10] & Ingunn M. Stromnes [1,2,4,5,10] ✉

T cell receptor (TCR) transgenic mice represent an invaluable tool to study antigen-specific immune responses. In the pre-existing models, a monoclonal TCR is driven by a non-physiologic promoter and randomly integrated into the genome. Here, we create a highly efficient methodology to develop T cell receptor exchange (TRex) mice, in which TCRs, specific to the self/tumor antigen mesothelin (*Msln*), are integrated into the *Trac* locus, with concomitant *Msln* disruption to circumvent T cell tolerance. We show that high affinity TRex thymocytes undergo all sequential stages of maturation, express the exogenous TCR at DN4, require MHC class I for positive selection and undergo negative selection only when both *Msln* alleles are present. By comparison of TCRs with the same specificity but varying affinity, we show that *Trac* targeting improves functional sensitivity of a lower affinity TCR and confers resistance to T cell functional loss. By generating P14 TRex mice with the same specificity as the widely used LCMV-P14 TCR transgenic mouse, we demonstrate increased avidity of *Trac*-targeted TCRs over transgenic TCRs, while preserving physiologic T cell development. Together, our results support that the TRex methodology is an advanced tool to study physiological antigen-specific T cell behavior.

The understanding of antigen-specific T cell responses at steady state and in disease have benefited from the use of T cell receptor (TCR) transgenic mice. TCR transgenic mice have a monoclonal TCR randomly integrated into the mouse genome and TCR expression is driven by heterologous promoter fragments including MHC class I[1], CD2[2,3], or endogenous promoter and regulatory flanking regions[4,5]. Such models require substantial time to generate, have random TCR

genomic integration, and the use of non-physiologic promoters may alter T cell functionality and hinder direct TCR comparisons.

Mesothelin (Msln) is a self-antigen overexpressed in many malignancies[6–11] and a promising target for cancer therapy[12]. Human MSLN-specific T cells are induced following vaccination demonstrating its immunogenicity[13]. Engineered T cells targeting MSLN are undergoing testing for the treatment of advanced malignancies[14–16]. Msln is

[1]Department of Microbiology and Immunology, University of Minnesota, Minneapolis, MN, USA. [2]Center for Immunology, University of Minnesota, Minneapolis, MN, USA. [3]Department of Pediatrics, University of Minnesota, Minneapolis, MN, USA. [4]Masonic Cancer Center, University of Minnesota, Minneapolis, MN, USA. [5]Center for Genome Engineering, University of Minnesota, Minneapolis, MN, USA. [6]Stem Cell Institute, University of Minnesota, Minneapolis, MN, USA. [7]Mouse Genetics Laboratory, University of Minnesota, Minneapolis, MN, USA. [8]Present address: Department of Medicine, UCLA Health, Los Angeles, CA, USA. [9]Deceased: Ebony A. Miller. [10]These authors jointly supervised this work: Beau R. Webber, Ingunn M. Stromnes. ✉e-mail: ingunn@umn.edu

lowly expressed in the pleura, peritoneum and pericardium in mice and humans, yet $Msln^{-/-}$ mice lack a discernible phenotype[17]. We previously characterized a panel of TCRs specific to Msln for pursuing a TCR engineered T cell therapy for cancer patient treatment[7]. We selected murine TCRs specific to $Msln_{406-414}$:H-2D$^b$ because T cells reactive to this epitope were obtainable from both wild type (WT) and $Msln^{-/-}$ mice, thereby modeling TCR affinity enhancement[18]. The adoptive transfer of P14 TCR transgenic T cells[1] retrovirally transduced to express a high affinity murine Msln-specific TCR (clone 1045) accumulated in malignant sites, elicited objective responses and prolonged survival in an autochthonous pancreatic ductal adenocarcinoma (PDA) mouse model[7]. The identical approach prolonged survival in the ID8 ovarian cancer model[19]. Finally, human CD8 T cells transduced to express MSLN-specific TCRs kill pancreatic[7] and ovarian[19] cell lines in vitro supporting the translation of this approach (NCT04809766). However, using the above approach, TCR engineered T cells were ultimately rendered dysfunctional in the highly suppressive tumor microenvironment (TME)[7]. Concomitant blockade of multiple immune checkpoints[20] or macrophage depletion[21] failed to rescue intratumoral engineered T cell dysfunction. Thus, additional strategies are necessary to enhance TCR engineered T cell therapy of cancer.

Prior studies have shown that targeting a chimeric antigen receptor (CAR)[22] or TCR[23] into the physiological *TRAC* locus in primary human T cells confers potent antigen-specific T cell function. Previously, a CRISPR ribonucleoprotein (RNP) electroporation and adeno-associated viral (AAV) donor infection (CRISPR-READI) zygote engineering approach was shown to be a robust method for targeted gene engineering[24]. Thus, we sought to combine the above approaches to generate physiologically regulated and naïve tumor-reactive T cells as a robust tool to elucidate how to enhance T cell antitumor function.

Here, we identify a robust and efficient methodology for direct orthotopic TCR exchange into the endogenous *Trac* locus in murine zygotes, referred to as T cell receptor exchange (TRex). First, we target a high or low affinity Msln-specific TCR to the *Trac* locus while simultaneously disrupting the cognate self-antigen Msln and assess T cell development and function. Next, we rapidly generate P14 TRex mice and compare T cell development and function to historical P14 TCR transgenic mice. In sum, we identify a highly efficient method to generate mice with physiological expression of a desired antigen-specific TCR, allowing for a standardized source of antigen-specific T cells. Further, our characterization of TRex mice reveals novel insights into T cell development and functionality.

## Results

### Targeting Msln-specific TCRs to *Trac* in primary murine T cells

We previously cloned and validated two murine $Msln_{406-414}$:H-2D$^b$-specific TCRs[7]. The 1045 TCR was the highest affinity obtained from $Msln^{-/-}$ mice and the 7431 TCR was the highest affinity TCR obtained from wild-type mice[7]. Both TCRs utilized Vα4 and Vβ9 and differed only in CDR3[7], which largely contributes to TCR specificity[25]. To develop an approach to target these TCRs to *Trac*, we first screened rAAV-GFP serotypes and identified rAAV6 to be superior at infecting activated primary mouse T cells without impacting viability (Supplementary Fig. 1a, b), similar to human T cells[26]. We designed and cloned 1045 or 7431 TCRβ variable (V), TCRβ constant (C) and TCRα V, linked by a self-cleaving P2A element[27] and flanked by homology arms (HA) encoding endogenous *Trac* genomic sequences into rAAV6 (Fig. 1a). Two single guide RNAs (sgRNAs) specific to the beginning of *Trac* exon 1 complexed to Cas9 ribonucleoprotein (RNP) were tested independently in activated polyclonal T cells (Fig. 1b), using an optimized CRISPR protocol[28]. Both *Trac* sgRNAs caused TCR and CD3 loss in >90% of T cells (Fig. 1c). While both sgRNAs reduced endogenous Vβ9, 1045 or 7431 TCR integrated only following *Trac* sgRNA 2 but not *Trac*

1 sgRNA (Fig. 1d, e). This is likely due to sgRNA 1 cutting within *Trac* exon 1 (Supplementary Fig. 1c, d). To assess TCR functionality, we restimulated the T cells with peptide-pulsed irradiated syngeneic splenocytes and quantified the frequency of Vβ9+ T cells 5 days later. Vβ9+ T cell frequency ranged from 5 to 10% prior to restimulation (Fig. 1g) and increased to 38–70% following cognate antigen stimulation (Fig. 1f), corresponding to a five-fold increase in T cell number. Thus, the targeted approach resulted in functional exogenous TCRs in primary T cells.

### Targeting TCRs into *Trac* sustains engineered T cell function and obviates Treg expansion

We next compared the efficiency of retroviral transduction (RV) of P14 T cells (Fig. 2a) using our prior protocol[7] to CRISPR/Cas9 + rAAV-mediated TCR *Trac* knock-in (KI) of polyclonal T cells (Fig. 2b) using the gating strategy in Supplementary Fig. 2a. Vβ9+ CD4 T cell frequency was comparable among RV and KI strategies whereas RV increased Vβ9+ CD8 T cell frequency (Fig. 2c–e). While proliferation was similar among the two approaches (Fig. 2f), RV increased Vβ9+ CD4+ Foxp3+ Treg frequency (Fig. 2g, h). Foxp3 was not detected in CD8 T cells as expected (Fig. 2g, i). We next measured cytokine production by RV and KI T cells following repetitive antigen stimulations. We found that Vβ9+ CD4 T cells rarely produced cytokines (Fig. 2j). In contrast, a higher frequency of RV Vβ9+ CD8 T cells produced cytokines following the second in vitro stimulation compared to KI Vβ9+ CD8 T cells (Fig. 2k). By the third stimulation, a greater frequency of KI Vβ9+ CD8 T cells produced cytokines compared to RV Vβ9+ CD8 T cells (Fig. 2k). Despite distinct cytokine production profiles, Vβ9 mean fluorescence intensity (MFI) was similar in KI vs. RV T cells (Supplementary Fig. 2b). Thus, the KI approach permitted TCR engineering of polyclonal T cells, obviated Treg expansion, and maintained cytokine production after recurrent antigenic exposure. Both approaches required the in vitro expansion of effector T cells for TCR integration, thereby prohibiting studies of naive Msln-specific T cells. Additionally, the low efficiency of TCR integration was a consideration of the KI approach .

### Rapid generation of Msln-specific TRex mice that lack *Msln*

To create a standardized and reproducible source of naïve murine Msln-specific T cells, we sought to create Msln-specific TRex mice. To circumvent potential T cell tolerance to Msln, we tested 2 *Msln* exon 4-specific sgRNAs to concurrently knockout Msln. Both sgRNAs efficiently induced insertion-deletion mutations (indels) in 3T3 fibroblasts (Supplementary Fig. 3a–e). Both 1045 and 7431 TCRs integrated in 40% of EL4 cells regardless of simultaneous *Msln* targeting (Fig. 3a, b). Thus, we next adapted the CRISPR-READI approach[24] to integrate the 1045 or 7431 TCRs (Supplementary Fig. 3f, g) in-frame with *Trac* while simultaneously disrupting *Msln* in murine zygotes (Fig. 3c). Using a junction PCR to detect exogenous TCR integration in one or both *Trac* alleles (Fig. 3d), 5/15 (33%) pups born were 1045 TCR heterozygous (1045$^{+/-}$, Fig. 3e). 1045$^{+/-}$ pups exhibited increased CD8 and decreased CD4 T cell percentages in circulation compared to wild type (WT) mice (Fig. 3f, g, Supplementary Fig. 3h), consistent with forced expression of an MHC I-restricted TCR. In 1045$^{+/-}$ pups, most CD8 T cells expressed Vβ9 and Vβ9 was also enriched in CD4 T cells (Fig. 3f, g and Table 1). Indels in *Msln* were detected in 9/15 animals, including all 1045$^{+/-}$ pups (Table 1).

We repeated this process with the lower affinity Msln-specific 7431 TCR and found 12/13 (92%) of the pups were 7431$^+$ with 6/13 (46%) homozygous and 6/13 (46%) heterozygous for the 7431 TCR (Fig. 3h). Circulating T cells were significantly biased toward CD8 T cells at the expense of CD4 T cells in 7431$^+$ pups (Fig. 3i-j, Supplementary Fig. 3h). In 7431$^{+/-}$ mouse #9, circulating T cells were markedly reduced, lacked Vβ9 (Fig. 3i), and contained an additional PCR band (Fig. 3h, Supplementary Fig. 3i) suggesting *Trac* was cut yet the TCR failed to integrate

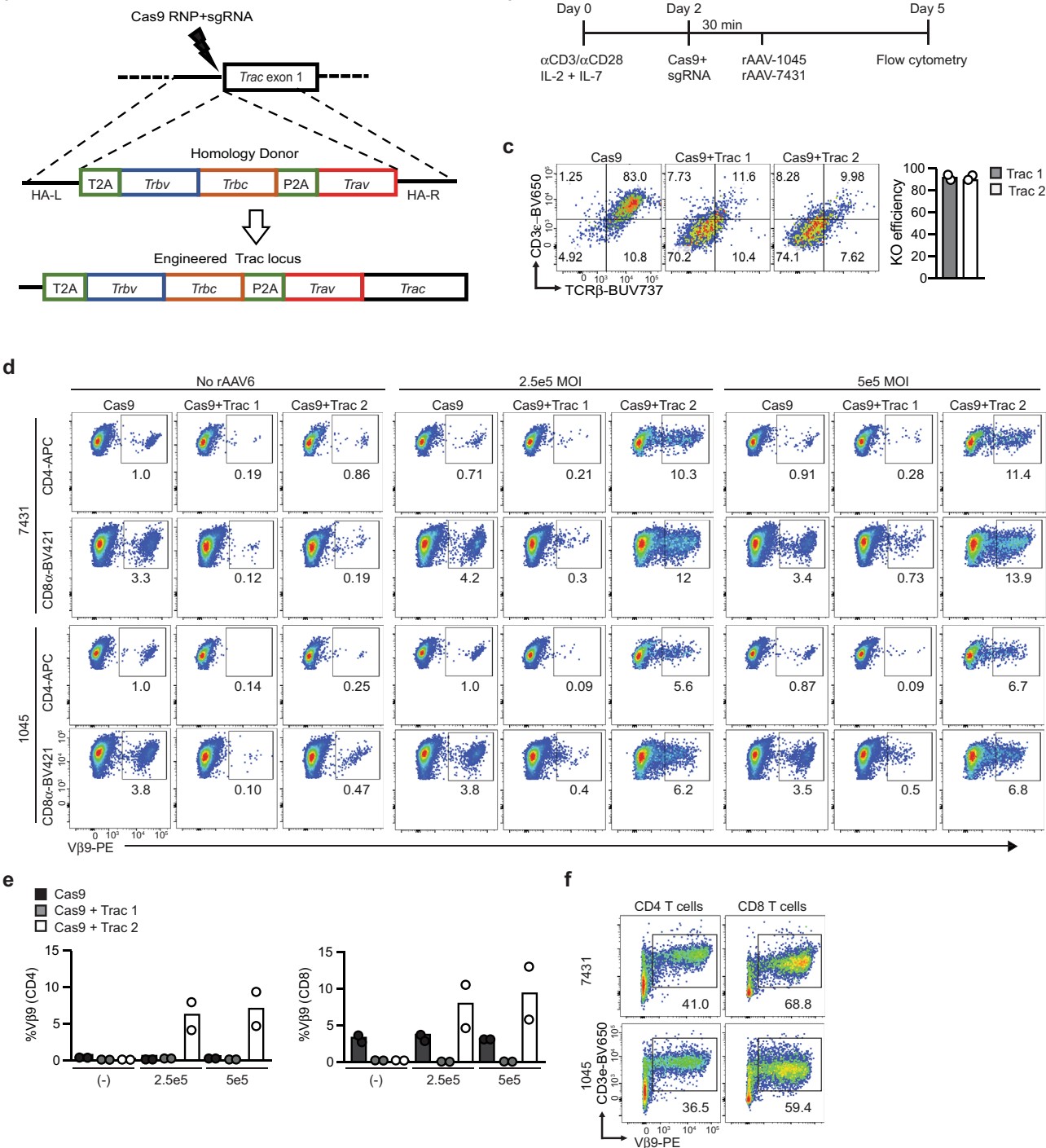

**Fig. 1 | Targeting Msln-specific TCRs to *Trac* in primary murine T cells.**
**a** Schematic of TCR targeting to murine *Trac*. sgRNAs targeting the region upstream of *Trac* exon 1 were designed. Codon optimized 1045 or 7431 TCRβ variable (V), TCRβ Constant (C) and TCRα V, linked by a self-cleaving P2A is flanked by left and right homology arms (HA-L, HA-R) and encoded by recombinant adeno-associated virus (rAAV) serotype 6. **b** Protocol for testing TCR targeting using CRISPR/Cas9 and rAAV. Polyclonal B6 splenocytes were activated with anti-CD3 + anti-CD28, IL-2 and IL-7. Two days after T cell activation, splenocytes were electroporated with Cas9 RNP complexed to *Trac*-specific sgRNAs, followed by

addition of rAAV encoding TCRs. **c** Knockout efficiency (KO) of *Trac* sgRNA 1 or 2 was measured by loss of TCRβ and CD3ε by flow cytometry on day 5 post activation. *n* = 2 biologically independent experiments. **d** Representative flow cytometry plots of 1045 or 7431 TCR expression in murine T cells was determined by Vβ9 staining on day 5 after varying the multiplicity of infection (MOI). **e** Proportion of CD4 or CD8 T cells that express Vβ9 at the indicated MOIs on day 5. Data are mean ± SEM and pooled from 2 independent experiments. **f** Frequency of donor TCR+ *Trac*-edited T cells 5 days post the second in vitro stimulation with Msln$_{406-414}$-pulsed irradiated splenocytes, IL-2 and IL-7.

correctly. In the remaining 7431⁺ pups, most T cells co-expressed CD3 and Vβ9 (Fig. 3i, j, Table 2) and exhibited *Msln* indels rates consistent with bi-allelic disruption (Table 2). Representative *Msln* sequence analysis is shown in Supplementary Fig. 3j–k. We detected Msln in WT

lung, but not in lungs from 7431 or 1045 TRex mice homozygous for *Msln* indels (e.g., *Msln$^{-/-}$*, Fig. 3k). Thus, we established an efficient method to target an exogenous TCR to *Trac* while disrupting a self/tumor-antigen in parallel.

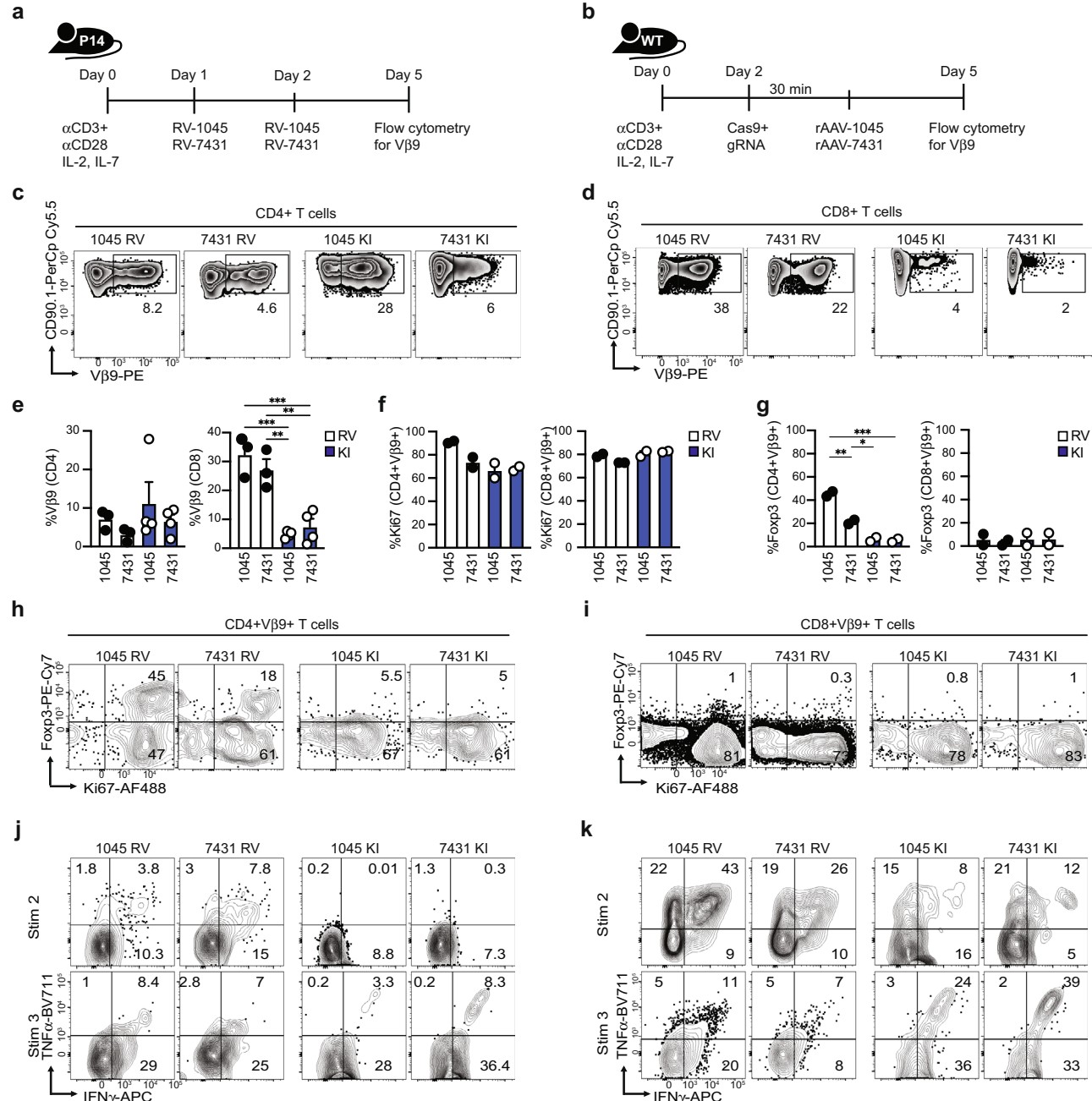

**Fig. 2 | Targeting TCRs to *Trac* sustains engineered T cell function and obviates Treg expansion. a** Schematic of retroviral transduction (RV) of P14 T cells with Msln-specific TCRs. **b** Schematic of CRISPR/Cas9 + rAAV TCR knockin (KI) of polyclonal T cells with Msln-specific TCRs. **c** Donor TCR expression by CD4 T cells 5 days post activation. **d** Donor TCR expression by CD8 T cells 5 days post activation. **e** Quantification of c and d. n = 2 biologically independent experiments. **f** Frequency of Ki67+ Vβ9+ CD4 or CD8 T cells 5 days post activation. n = 2 biologically independent experiments. **g** Frequency of Foxp3+ Vβ9+ CD4 or CD8 T cells 5 days post activation. *n* = 2 biologically independent experiments.

**h** Representative plots gated on live CD4+ Vβ9+ T cells. **i** Representative plots gated on live CD8+ Vβ9+ T cells. **j** Representative plots gated on CD4+ Vβ9+ T cells. Intracellular cytokine staining was assessed after the second (Stim 2) or third (Stim 3) restimulation in vitro with Msln peptide-pulsed irradiated syngeneic splenocytes and IL-2 + IL-7. **k** Representative plots gated on CD8+ Vβ9+ T cells. Intracellular cytokine staining was assessed as in **j**. Data are mean ± SEM in e–g. *n* = 3–6 mice per group. *$p < 0.05$, **$p < 0.005$, and ***$p < 0.0005$. One-way ANOVA with a Tukey's post test.

## High affinity Msln-specific T cells undergo central tolerance in a *Msln* dose dependent manner

To investigate T cell development in TRex mice, we backcrossed 1045 TRex #11 (Fig. 3e, Table 1) onto *Msln^{WT/WT}*, *Msln^{WT/−23}* and *Msln^{−23/−23}* background (referred to as *Msln^{+/+}*, *Msln^{+/−}*, and *Msln^{−/−}*, respectively, Table 1) and analyzed thymocytes in 1045^{+/+} offspring. Thymus weight (Fig. 4a) and CD45+ cell number (Supplementary Fig. 4a) were similar regardless of *Msln* status. Thymocyte maturation occurs through

sequential differentiation program that is distinguished by CD4 and/or CD8 coreceptor expression[29]. The earliest thymocyte progenitors lack CD4 and CD8 (double negative, DN) and differentiate into CD4+ CD8+ double positive (DP) followed by maturation into CD4 or CD8 single positive (SP) cells. The frequency and number of DNs and DPs were similar among the strains (Fig. 4b–d). In 1045^{+/+} *Msln^{+/−}* and *Msln^{−/−}* TRex mice, thymocytes were biased toward CD8 SP (Fig. 4b–d). In contrast, CD8 SP frequency and number were significantly reduced

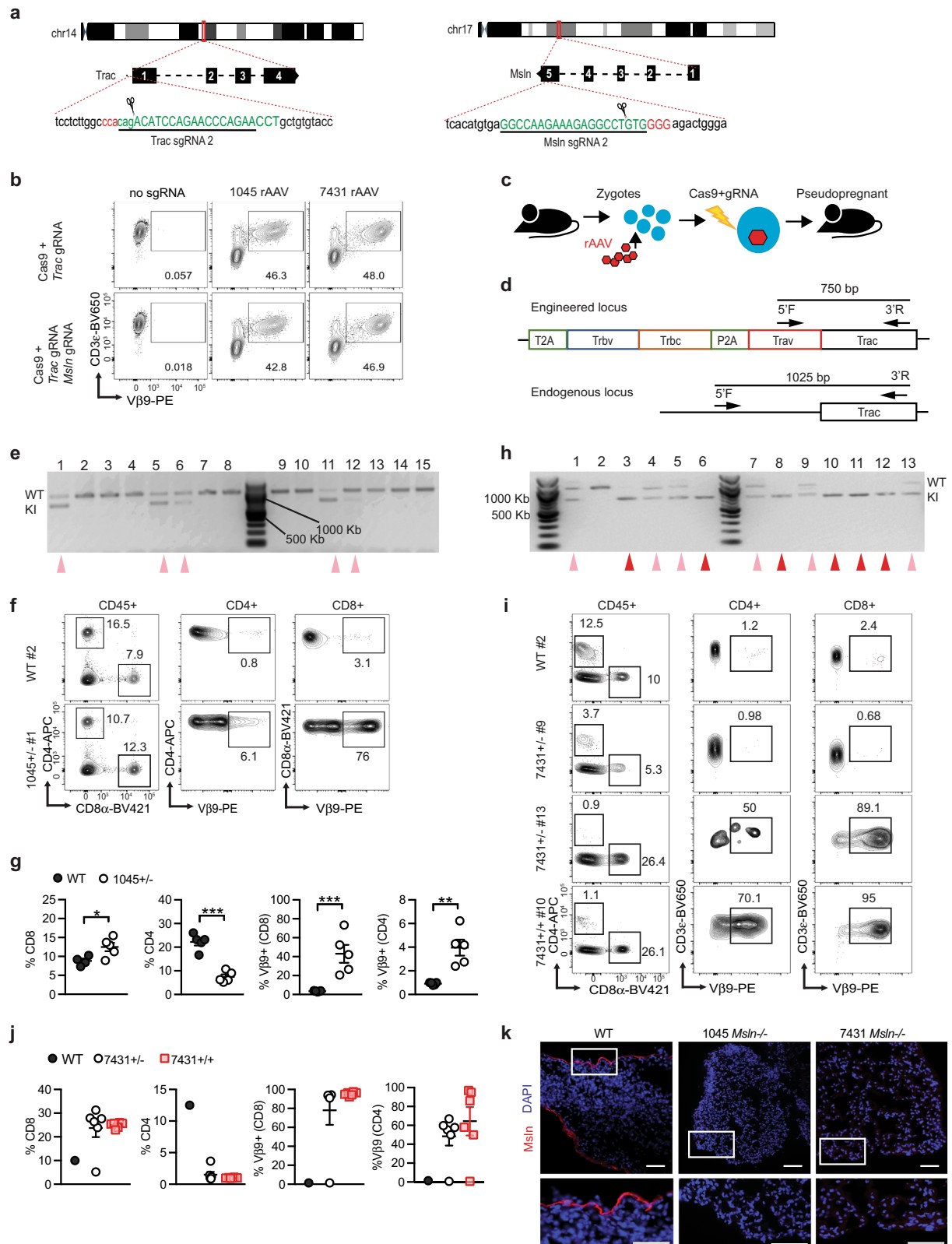

in *Msln*[+/+] as compared to *Msln*[+/−] and *Msln*[−/−] 1045 TRex mice (Fig. 4c, d, Supplementary Fig. 4b). These data suggest negative selection to *Msln* is gene dose dependent. Vβ9 was increased in most thymocyte stages in TRex vs. WT mice as expected (Fig. 4e, f) and Vβ9+ thymocytes downregulated CD24, consistent with maturation (Fig. 4e). Vβ9+ DP and Vβ9+ CD8 SP numbers trended to be reduced in *Msln*[+/+] as

compared to *Msln*[+/−] and *Msln*[−/−] 1045[+/+] TRex mice (Fig. 4g), further supporting a gene dosage dependent induction of central tolerance.

The DN stage is further subdivided into DN1- DN4 based on CD25 and CD44 (Supplementary Fig. 4c)[30]. TCRβ is rearranged first at DN3[29]. Rapid cell proliferation and TCRα upregulation occurs in the transition from DN4 to DP stage and selection ensues for functional αβ TCR

**Fig. 3 | Rapid generation of Msln-specific TRex mice that lack *Msln*. a** Sequence and target sites of sgRNAs targeting *Trac* exon 1 or *Msln* exon 4. **b** CD3+ EL4 cells were electroporated with *Trac* sgRNA or *Trac* and Msln sgRNAs complexed with Cas9 RNP, followed by rAAV-1045 or rAAV-7431. Msln TCR expression was determined by Vβ9. **c** Schematic for murine zygote engineering. **d** Junction PCR design to detect TCR integration. 5′ Forward primers are located within the engineered locus to detect the exogenous TCR and in an intron region upstream of the endogenous *Trac* locus to detect the WT allele. A 3′ Reverse primer was located in *Trac*. **e** PCR gel of junction PCR performed with DNA collected from the 15 pups born from 1045 engineered pseudo pregnant B6 females. 100 Kb ladder is located in the middle lane and 500 and 1000 Kb demarcated. Wild type (WT) indicates amplification of the endogenous locus while knock-in (KI) indicates amplification of the exogenous TCR upstream of the *Trac*. Pink arrows, 1045⁺/⁻ heterozygous pups. n = 1

experiment. **f** T cell frequency in circulation (left, gated on live CD45+ cells) and frequency of Vβ9+ CD4 (middle) or CD8 (right) T cells in representative pups from **e**. **g** Quantification of **f**. Each dot is an independent mouse. n = 5 mice per group. Data are mean ± SEM. **h** Junction PCR from DNA isolated from the 13 pups born from the 7431-zygote engineered mice. 100 Kb ladder is located on left and middle lanes with 500 and 1000 Kb demarcated. Pink arrows indicate 7431⁺/⁻ pups and red arrows indicate 7431⁺/⁺ pups. n = 1 experiment. **i** T cell frequency in circulation (left, gated on live CD45+ cells) and frequency of CD4 (middle) and CD8 (right) T cells expressing Vβ9 in representative pups from **h**. **j** Quantification of **i**. Each dot is an independent mouse and include n = 13 animals from **h**. Data are mean ± SEM. **k** Representative immunofluorescent (IF) tissue staining of Msln (red) and DAPI (blue) in lungs from WT, 1045 *Msln*⁻/⁻, and 7431 *Msln*⁻/⁻ mice. Scale bar, 25 µM. Representative of six biologically independent animals.

**Table 1 | Analysis of T cells and *Msln* locus from 1045 TRex pups**

| # | ID | Sex | 1045ᵃ | %CD8ᵇ | %Vβ9 (CD8) | %CD4 | %Vβ9 (CD4) | %*Msln* KOᶜ,ᵇ | *Msln* In/Dels |
|---|------|-----|-------|-------|------------|-------|------------|----------------|----------------|
| 1 | 3290 | M | +/− | 13 | 73.3 | 10.7 | 6.22 | 84% | +1/−7 |
| 2 | 3291 | M | −/− | n.d. | n.d. | n.d. | n.d. | 50% | −3/−8 |
| 3 | 3292 | M | −/− | n.d. | n.d. | n.d. | n.d. | 0% | none |
| 4 | 3293 | M | −/− | n.d. | n.d. | n.d. | n.d. | 11% | +1 |
| 5 | 3294 | M | +/− | 13.7 | 52.4 | 6.09 | 4.19 | 93% | +4/+1/−1/−7 |
| 6 | 3295 | M | +/− | 11.3 | 26.7 | 5.23 | 2.75 | 10–50% | n.r. |
| 7 | 3296 | F | −/− | n.d. | n.d. | n.d. | n.d. | 0% | none |
| 8 | 3297 | F | −/− | n.d. | n.d. | n.d. | n.d. | 90% | +13/+1 |
| 9 | 3298 | F | −/− | n.d. | n.d. | n.d. | n.d. | 0% | none |
| 10 | 3299 | F | −/− | n.d. | n.d. | n.d. | n.d. | 0% | none |
| 11 | 3300 | F | +/− | 15.5 | 20.5 | 9.04 | 4.29 | 93% | −23 |
| 12 | 3301 | M | +/− | 8.91 | 42.3 | 6.29 | 2.39 | 40–50% | +1/−1/−2 |
| 13 | 3302 | M | −/− | n.d. | n.d. | n.d. | n.d. | n.r. | n.r. |
| 14 | 3303 | M | −/− | n.d. | n.d. | n.d. | n.d. | 45% | −1/−2 |
| 15 | 3304 | M | −/− | n.d. | n.d. | n.d. | n.d. | 47% | +1 |

ᵃ1045 *Trac* knock-in was determined by junction PCR from tail DNA.
ᵇn.d., not determined; n.r., no results indicating sequence analysis was attempted but data were inconclusive.
ᶜMsln knockout was determined by PCR amplification of *Msln* exon 4 followed by Sanger sequencing and Inference of Crispr Edits (ICE) analysis (https://www.synthego.com/products/bioinformatics/crispr-analysis).

heterodimers on DP cells[31]. Since the 1045 TCR is integrated in *Trac*, it is expected that the donor TCR would be detectable at the DN4 stage. As such, Vβ9 was first detected at the DN4 stage cells in 1045 TRex mice (Fig. 4h), supporting physiological TCR regulation and thymocyte maturation in TRex mice.

To assess if MHC I is required for T cell development, we backcrossed 1045⁺/⁺ TRex mice to the *B2m*⁻/⁻ mice, which lack functional MHC I[32]. Thymus weight trended to be smaller in 1045⁺/⁺ *B2m*⁻/⁻ vs. *B2m*⁺/⁺ mice (Fig. 4i). CD8 SP frequency was dramatically reduced in 1045⁺/⁺ *B2m*⁻/⁻ vs. *B2m*⁺/⁺ TRex mice and a compensatory increase in DP frequency in 1045⁺/⁺ *B2m*⁻/⁻ mice was detected (Fig. 4j-k). CD8+ Vβ9+ and CD4+ Vβ9+ T cell frequencies were reduced in 1045⁺/⁺ *B2m*⁻/⁻ TRex mice (Fig. 4l). Thus, MHC I is required for positive selection of TRex T cells.

**Peripheral 1045 T cells from *Msln*⁻/⁻ or *Msln*⁺/⁻ mice respond to specific antigen**

We next tested the functionality of peripheral T cells isolated from 1045 *Msln*⁺/⁻ and *Msln*⁻/⁻ mice. Most splenic T cells expressed Vβ9 in 1045⁺/⁻ or 1045⁺/⁺ mice (Fig. 5a–d). A higher frequency of CD8 (Fig. 5c) and CD4 (Fig. 5d) T cells expressed Vβ9 in 1045⁺/⁺ vs. 1045⁺/⁻ mice. Most splenic CD8+ Vβ9+ T cells exhibited a CD44ˡᵒʷCD62L⁺ naive phenotype in *Msln*⁻/⁻ and *Msln*⁺/⁻ 1045 TRex mice (Fig. 5c) consistent with lack of self-antigen recognition when one *Msln* allele is absent. In contrast, a higher frequency of CD4+ Vβ9+ T cells upregulated CD44 and downregulated CD62L in 1045⁺/⁺ vs. 1045⁺/⁻

*Msln*⁻/⁻ mice (Fig. 5d), a phenotype that was independent of self-antigen recognition.

To determine if 1045 T cells responded to specific antigen, we labeled splenocytes with a proliferation dye, incubated with Msln₄₀₆₋₄₁₄ and quantified T cell proliferation and activation after 3 days. We found that splenic CD8+ Vβ9+ T cells proliferated and upregulated CD25, CD69 and CD44 in response to Msln₄₀₆₋₄₁₄-pulsed APCs (Fig. 5e, f). In contrast, rare CD8+ Vβ9− T cells from the same TRex mice failed to respond to antigen (Fig. 5e). Presence of a single Msln allele did not impact peripheral 1045 T cell functionality in vitro (Fig. 5e, f). CD4+ Vβ9+ T cells from 1045 TRex mice upregulated CD69 and CD25 following antigen recognition, yet proliferation was modest and CD44 was not further increased consistent with suboptimal CD4 T cell activation (Fig. 5g, h).

As Tregs accumulated following P14 transgenic (Tg) T cell in vitro expansion (Fig. 2g, h), we compared Tregs from P14 Tg mice to 1045 and 7431 *Msln*⁻/⁻ TRex strains. Tregs were disproportionately enriched among CD4 T cells from P14 Tg compared to WT or TRex mice (Supplementary Fig. 5a–b). Tregs were biased toward a CD25- Foxp3+ subset in P14 mice (Supplementary Fig. 5a–b), potentially precursors to CD25+ Foxp3 + Treg[33,34], and were more proliferative (Supplementary Fig. 5c–d). In contrast to T cells from WT mice, T cells from P14 transgenic mice activated with αCD3 + αCD28 and IL-2 exhibited increased frequency of Foxp3+ CD25+ Tregs (Supplementary Fig. 5e–f), and many did not express Vα2, the P14 TCRα chain (Supplementary Fig. 5g). Thus, endogenous TCRα expression may be a

**Table 2 | Analysis of T cells and *Msln* locus from 7431 TRex pups**

| # | ID | Sex | 7431[a] | %CD8[b] | %Vβ9 (CD8) | %CD4 | %Vβ9 (CD4) | %Msln *Indel*[c,b] | *Msln* Indel |
|---|-----|-----|--------|---------|-----------|------|-----------|------------------|--------------|
| 1 | 3291 | F | +/− | 27.7 | 92.3 | 0.87 | 55.3 | 86[d] | −2,+4,−19 |
| 2[e] | 3292 | F | −/− | 9.98 | 2.31 | 12.5 | 1.24 | 0 | none |
| 3 | 3293 | F | +/+ | 26.1 | 94.9 | 1.11 | 70.1 | 97[d] | −7 |
| 4 | 3294 | F | +/− | 31.3 | 92.6 | 1.0 | 59.8 | 97[d] | −4 |
| 5 | 3295 | F | +/− | 23.9 | 91.7 | 1.4 | 67 | 84[d] | −8 |
| 6 | 3296 | M | +/+ | 25.7 | 74.4 | 1.1 | 95.5 | 91[d] | −5,+1 |
| 7 | 3297 | M | +/− | 27.9 | 91.6 | 1.9 | 57.9 | 95 (KO 47) | −7,−9 |
| 8 | 3298 | M | +/+ | 24.1 | 91 | 1.0 | 64.2 | 95[d] | −1,−8 |
| 9 | 3300 | M | +/−[e] | 5.25 | 0.98 | 3.67 | 0.68 | 86[d] | +1,−8 |
| 10 | 3301 | M | +/+ | 25.4 | 93.7 | 0.97 | 69 | 97[d] | −8 |
| 11 | 3302 | M | +/+ | 25.7 | 96.2 | 1.0 | 73 | 92 (KO 45) | −1,−6 |
| 12 | 3303 | M | +/+ | 22.0 | 92.8 | 0.96 | 73.5 | 96[d] | +1 |
| 13 | 3304 | M | +/− | 26.4 | 88.8 | 0.89 | 50 | 97[d] | −4 |

[a]7431 *Trac* knock-in was determined by junction PCR from tail DNA.

[b]n.d., not determined; n.r., no results indicating sequence analysis was attempted but data were inconclusive.

[c]Msln knockout was determined PCR amplification of *Msln* exon 4 followed by Sanger sequencing and Inference of Crispr Edits (ICE) analysis (https://www.synthego.com/products/bioinformatics/crispr-analysis).

[d]Knockout score was identical to % Indel.

[e]An additional band was detected between the WT and KI.

prerequisite, but not the only factor contributing to disproportionate Treg expansion in P14 transgenic mice. In addition, a higher proportion of CD4 T cells were Treg in OT1 TCR transgenic[35] compared to WT mice (Supplementary Fig. 5h). Thus, the TRex approach may overcome Treg abnormalities in traditional TCR transgenics.

## TCR *Trac* targeting improves the functional avidity of a low affinity TCR

We next compared the functionality of 7431[+/+] and 1045[+/+] T cells from *Msln*[−/−] TRex animals. Spleen weight (Supplementary Fig. 6a), CD45+ cell number (Supplementary Fig. 6b), and a bias toward the CD8 lineage (Fig. 6a) were similar among the two strains. While splenic CD8 T cell number was similar among the two TRex strains (Supplementary Fig. 6c), 1045[+/+] *Msln*[−/−] mice exhibited increased splenic CD4 T cell frequency (Fig. 6a) and number (Supplementary Fig. 6d). Over 95% of splenic CD8 T cells expressed Vβ9 (Fig. 6b) and exhibited a naive (CD44-CD62L+) phenotype (Fig. 6c) in both TRex strains. ViSNE analysis[36], which reduces high-parameter data into two dimensions for visualization, confirmed a resting (CD25-Ki67-) CD8 T cell phenotype (Supplementary Fig. 6e).

Peptide:MHC tetramers can be a proxy for TCR affinity and functional avidity[7,37–39]. While naïve 7431 and 1045 CD8 T cells stained similarly for Vβ9, 1045 CD8 T cells stained brighter for Msln$_{406-414}$:H-2D[b] tetramer (Fig. 6d), supporting that the 1045 TCR is higher affinity and consistent with prior analysis[7]. Following in vitro activation with antigen and IL-2, TRex T cells upregulated CD44, yet maintained CD62L indicative of antigen experience and maintenance of memory potential (Supplementary Fig. 6f). Activated 1045 T cells stained brighter for tetramer (Fig. 6d, e) and T cells with the highest tetramer staining were also brightest for CD25 and CD69 (Fig. 6e). Thus, the TRex methodology still translates into differences in tetramer staining.

T cells with high functional avidity respond to lower antigen concentrations which often correlate with effector T cell capacity[40]. Therefore, we compared 1045 and 7431 effector T cell cytokine production in response to titrating concentrations of antigen. Unexpectedly, the functional avidity between 1045 and 7431 TRex effector T cells was similar (Fig. 6f). These results contrast with our prior analysis of 1045 and 7431 RV T cells in which 1045 RV T cells produce IFNγ with a log-fold greater sensitivity as compared to 7431 RV T cells[7]. While 1045 effector T cells produced more IFNγ and TNFα on a per cell basis in response to high antigen concentrations, both TCRs elicited similar cytokine amounts in response to low antigen concentrations (Fig. 6g). Thus, despite decreased tetramer staining consistent with a lower affinity TCR, 7431 TRex effector T cells exhibit a functional avidity comparable to 1045 TRex effector T cells suggesting that integration at *Trac* may enhance antigen sensitivity.

TCR downregulation after antigen recognition may be a protective mechanism to prevent pathology by autoreactive CD8 T cells[41,42]. Directing a CAR to the *TRAC* locus in human T cells promotes CAR internalization and re-expression, which delays effector T cell differentiation and acquisition of an exhausted phenotype[22]. Targeting TCRs to *TRAC* also conferred productive antitumor human T cells[43]. As such, we next compared TCR downregulation in TRex effector T cells 5 h following a second antigenic stimulation. 1045 effector T cells downregulated Vβ9 to a greater extent than 7431 effector T cells (Fig. 6h, Supplementary Fig. 6g) whereas CD8α was similarly downregulated (Fig. 6i, Supplementary Fig. 6g). In addition, TCR downregulation was more pronounced in 1045 vs. 7431 T cells at multiple timepoints (Fig. 6j, k). 1045 TRex T cells exhibited higher and prolonged CD25 and PD-1 as compared to 7431 TRex T cells consistent with stronger TCR signaling (Fig. 6j, k).

High affinity MHC I-restricted TCRs can engage CD4 T cells due to peptide:MHC binding independent of the CD8 coreceptor[44]. Strikingly, over 90% of CD4 T cells were Vβ9+ in 1045 and 7431 TRex mice (Supplementary Fig. 6h). CD4 T cells stained brighter for tetramer in 1045 vs. 7431 TRex mice, despite comparable Vβ9 (Fig. 6l). A higher frequency of effector CD4 T cells produced cytokines in response to Msln peptide in 1045 vs. 7431 TRex mice (Fig. 6m). IFNγ produced per cell was elevated in 1045 vs. 7431 CD4 T cells (Fig. 6n). Thus, the MHC I-restricted high affinity 1045 TCR can elicit some function in CD4 T cells.

## Characterization of T cell development in P14 TRex mice

To identify differences between the TRex approach and historical TCR transgenics, we generated P14 TRex mice. We first validated this approach by targeting the P14 TCR to *Trac* in EL4 cells (Fig. 7a, b). We next repeated the protocol established in 1045 and 7431 TRex mice for the P14 TCR in murine zygotes and identified P14 TCR integration at the *Trac* locus in pups by junction PCR (Fig. 7b). 19/52 pups (37%) showed P14 TCR integration in at least one *Trac* allele (Fig. 7b, c, Supplementary Table 1), a frequency within the range of 1045 and 7431 TRex pups (Fig. 7c). Circulating T cell proportions were biased toward

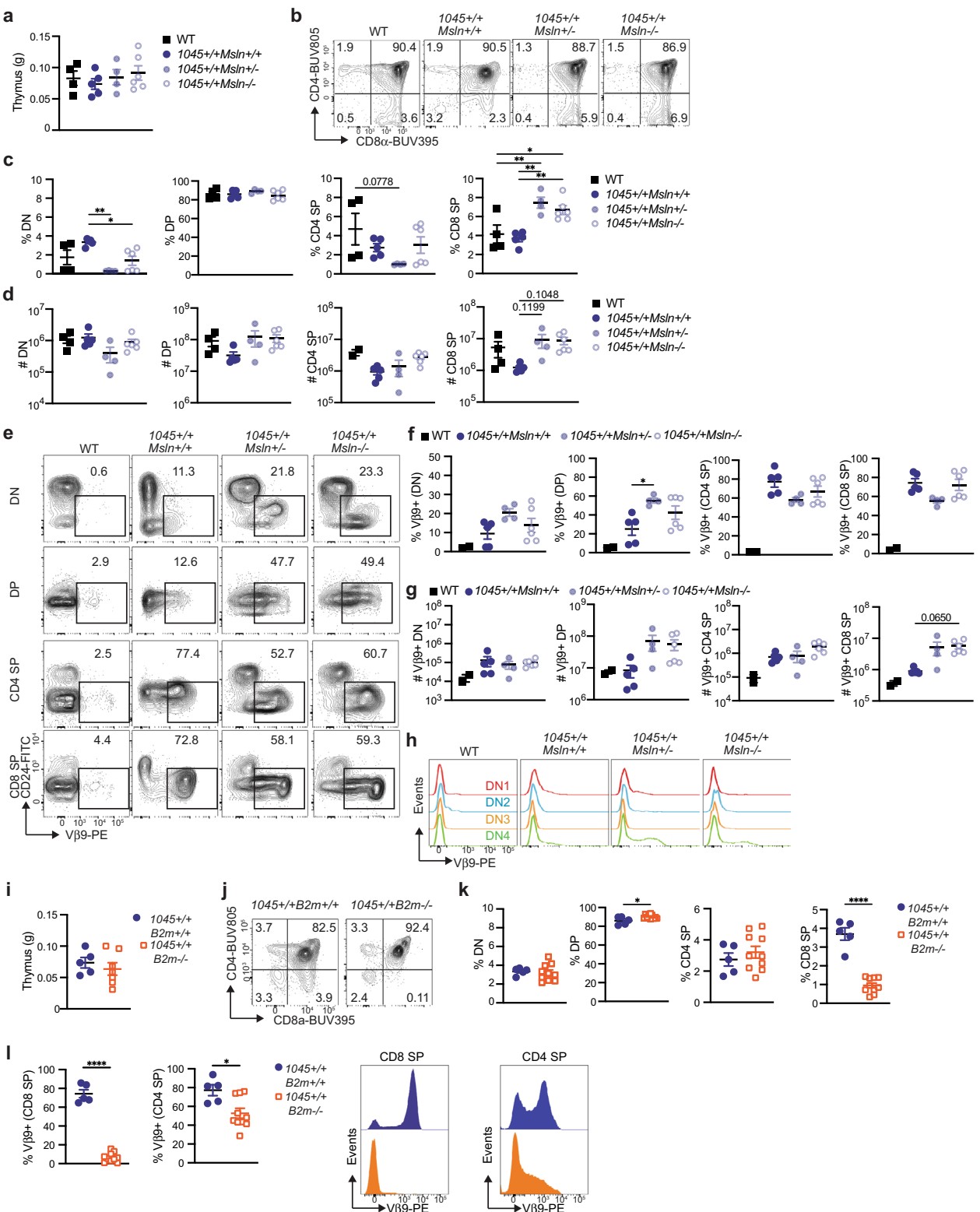

the CD8 lineage and co-expressed Vα2 and Vβ8 in both P14+/− and P14+/+ TRex pups (Fig. 7d, Supplementary Table 1). Thymus weight and immune cell number were similar among the strains (Fig. 7e). DN thymocyte frequency was increased in P14 Tg and was similarly low in P14 TRex and WT mice (Fig. 7f–h). DP and CD4 SP frequency was decreased and CD8 SP frequency was increased in both P14 Tg and TRex mice compared to WT mice (Fig. 7f–h). CD4 SP number was decreased in P14 Tg compared to WT whereas CD8 SP number was similarly increased in both P14 strains (Fig. 7g). P14 Tg thymus

contained increased DN4 and decreased DN1-DN3 frequency as compared to P14 TRex and WT thymus (Fig. 7i, j). While DN P14 Tg T cells were enriched for Vα2+ Vβ8+ cells, TRex T cells became enriched for Vα2+ Vβ8+ at later thymocyte development stages (Fig. 7k, l). Vβ8 was first detected at DN1 in P14 Tg thymocytes but at the DN4 stage in P14 TRex thymocytes (Fig. 7m, Supplementary Fig. 7a), the latter consistent with the known timing of TCRα expression in WT thymocytes. Only 20% of CD4 SP co-expressed Vα2 and Vβ8 in P14 Tg whereas over 90% of CD4 SP expressed the P14 TCR in TRex mice (Fig. 7k). Most CD8

**Fig. 4 | High affinity Msln-specific T cells undergo central tolerance in a *Msln* dose dependent manner. a** Thymus weight in grams (g). $n = 4$–6 biologically independent animals. Data are mean ± SEM. **b** Representative plots gated on live CD45+ B220− thymocytes. **c** Frequency double negative (DN), double positive (DP), CD4 single positive (CD4 SP) and CD8 single positive (CD8 SP) among CD45+ B220− thymocytes. $n = 4$–6 biologically independent animals. Data are mean ± SEM. $p < 0.05$, $**p < 0.005$. One-way ANOVA with a Tukey's posttest. **d** Mononuclear CD45+ cell number per thymus. Each dot is an independent mouse. Data are mean ± SEM. $n = 4$–6 mice per group. $*p < 0.05$, $**p < 0.005$. Anova with a Tukey's posttest. **e** Representative plots of Vβ9 and CD24 of thymocytes in the indicated development stage. Numbers in plots indicate the frequency of Vβ9+ cells. **f** Vβ9+ cell frequency in each thymocyte developmental stage. $n = 4$–6 biologically independent animals in 1045 cohorts and $n = 2$ biologically independent animals in the WT group. Data are mean ± SEM. $*p < 0.05$. One-way ANOVA with a Tukey's posttest. **g** Number of Vβ9+ cells at each developmental stage per thymus. Each dot is an independent mouse. $n = 4$–6 biologically independent animals in 1045 groups and $n = 2$ biologically independent animals in the WT group. Data are mean ± SEM. One-way ANOVA with a Tukey's posttest. **h** Representative Vβ9 staining of thymocytes in DN1-DN4. **i** Thymus weight. Data are mean ± SEM. $n = 5$–10 biologically independent animals. **j** Representative plots gated on live CD45+ B220- thymocytes. **k** Frequency of cells in each thymocyte stage. Data are mean ± SEM. $n = 5$–10 biologically independent animals. $*p < 0.05$, $****p < 0.0001$. Two-tailed unpaired Student's $T$ test. **l** Vβ9 frequency among CD8 SP and CD4 SP thymocytes (top) and representative histograms (bottom). Quantified data are mean ± SEM. $n = 5$–10 biologically independent animals. $*p < 0.05$, $****p < 0.0001$. Two-tailed unpaired Student's $T$ test.

SP in TRex mice were CD3+ Vα2+ Vβ8+ (Fig. 7k, l, Supplementary Fig. 7b–c). CD3ε intensity was lower in CD4 SP than CD8 SP in P14 TRex mice and higher in CD8 SP in TRex than Tg and WT mice (Supplementary Fig. 7d).

To assess endogenous Vβ in TRex mice, we stained thymocytes with a pool of FITC-conjugated antibodies specific to multiple murine Vβ alleles while excluding Vβ8, which was detected using a different fluorochrome using the gating strategy in Supplementary Fig. 7e. The Vβ panel detected 40–60% of endogenous Vβs in WT CD3+ thymocytes (Fig. 7n), an expected range since there are ~21 functional Vβ genes in mice[45]. A fraction of CD3+ thymocytes co-expressed low levels of an endogenous Vβ with Vβ8 in TRex mice (Fig. 7n–p). Endogenous Vβ was significantly decreased in TRex CD8 SP as compared to WT CD8 SP (Supplementary Fig. 7f). P14 Tg and TRex DN4 thymocytes exhibited slightly increased dual Vβ frequencies compared to WT mice (Supplementary Fig. 7g). To investigate the frequency of dual Vβ T cells in circulation, we performed a similar analysis of T cells in the peripheral blood. Approximately, 20% of peripheral CD8, CD4 conventional (Tcon) and CD4+ Foxp3+ cells co-expressed an endogenous and exogenous Vβ in P14 TRex mice (Fig. 7p, Supplementary Fig. 7g–i). In 1045$^{+/+}$ *Msln*$^{−/−}$ TRex mice, 30–40% of peripheral CD8, CD4 Tcon, and Treg were dual Vβ+ (Supplementary Fig. 7j) revealing some interstrain variability in endogenous Vβ. To further investigate if thymocyte maturation is physiological in P14 TRex mice, we evaluated CD69 as it is downregulated in mature SP thymocytes. CD69- CD8 SP thymocytes also expressed CD62L (Supplementary Fig. 7k), consistent with maturation. CD69− (mature) CD3+ CD8 SP thymocyte frequency was decreased in TRex mice as compared to P14 Tg and WT mice (Fig. 7q, Supplementary Fig. 7l). However, P14 Tg and TRex mice exhibited a 3-fold increase in mature CD3+ CD8 SP number as compared to WT mice (Fig. 7r), indicating CD8 SP maturation is intact in TRex mice.

### Targeting a TCR to *Trac* enhances exogenous TCR expression and antigen sensitivity

We next compared peripheral T cell responses between P14 TRex and P14 Tg mice to precisely uncover the role of the genomic location on T cell function. Spleen weights were similar among the strains (Supplementary Fig. 8a). CD8 T cell frequency and number were increased in both P14 strains as compared to WT mice (Fig. 8a, Supplementary Fig. 8b). CD8 T cells equally co-expressed Vα2 and Vβ8 among the two P14 strains (Fig. 8b), whereas more CD4 T cells were Vα2+ Vβ8+ in TRex vs. Tg mice (Fig. 8b–c). Most CD8 T cells exhibited a naïve phenotype whereas most CD4 T cells expressed CD44 in both strains (Supplementary Fig. 8c). TRex T cells expressed more CD3ε, Vα2, and Vβ8 ex vivo (day 0) and following activation (day 6) than analogous P14 Tg T cells (Fig. 8d, Supplementary Fig. 8d). CD8 MFI was comparable ex vivo and modestly higher after activation in TRex T cells (Supplementary Fig. 8d). CD25 was also higher in CD8 P14 TRex than Tg effector T cells (Fig. 8d). After primary activation, the kinetics of TCR internalization and re-expression were similar in TRex and Tg T cells (Fig. 8e), which suggests that factors other than the genomic location

impact TCR downregulation. We next compared proliferation by incubating CTV-labeled splenocytes with titrating concentrations of antigen and IL-2. At low antigen concentrations, TRex T cells were slightly more proliferative (Fig. 8f) and maintained higher TCR levels than Tg T cells (Fig. 8f). Providing exogenous IL-2 may compensate for differences in TCR signaling and proliferation, therefore we repeated the proliferation assay without IL-2. A greater frequency of TRex T cells were proliferating at low antigen concentration (Fig. 8g) corresponding to CD69 upregulation (Fig. 8g) and CD44, whereas PD-1 was not affected (Supplementary Fig 8e). Further, more TRex T cells had undergone ≥ 3 cell divisions (Fig. 8h) and trended to produce more IFNγ than analogous Tg T cells, particularly noticeable at lower antigen concentrations (Fig. 8i). CD69 MFI and frequency of cells expressing CD25 were also greater in TRex vs. Tg effector T cells (Fig. 8i). Thus, the data suggest that targeting a TCR to *Trac* may improve T cell responsiveness to antigen.

### Discussion

We develop a methodology to integrate a desired exogenous TCR into the physiological endogenous *Trac* locus in murine T cells and zygotes. We adapted the micro-injection free CRISPR-READI approach[24] by combining AAV-mediated donor TCR delivery with Cas9/sgRNA RNP electroporation to induce site-specific modifications in *Trac* and integrate the exogenous TCR. Contemporaneously, we induced *Msln* null mutations to circumvent T cell tolerance. We independently generate multiple TRex strains supporting high efficiency and reproducibility of this method. As Msln is overexpressed by many solid tumors[6–11], the 1045 and 7431 TRex mice will provide a standardized source of naive T cells with physiological expression of TCRs specific to a native and clinically relevant tumor antigen to identify how to sustain T cell function in solid tumors.

TCR transgenic mice have improved our understanding of T cell development and differentiation. Considerations of this approach include multiple and random TCR integration into the genome resulting in non-physiologic TCR regulation[1–5] and premature TCRα and TCRβ expression at the DN1 stage impacting thymocyte development[46]. TCR rearrangement is a highly ordered and sequential process where TCRβ is rearranged in DN3 preceding TCRα rearrangement that can be initiated at DN4 transitioning to the DP stage. A productive TCRβ rearrangement prevents further Vβ-to-DJβ rearrangements at the DP stage, a process called allelic exclusion[45]. As a dsDNA break is directly upstream of *Trac* resulting in endogenous TCRα loss in TRex mice, exogenous TCR integration is critical for T cell development in TRex mice. We further show that exogenous TCRα and TCRβ are initially expressed at DN4 and thymocytes undergo all the sequential stages of maturation in TRex mice. Indeed, MHC I is required for TRex T cell positive selection and negative selection of TRex T cells can occur when the target native antigen is sufficiently expressed. One consideration of the TRex approach is a fraction of TRex T cells express endogenous TCRβ in addition to the exogenous TCRβ. This is expected as the endogenous *Tcrb* locus remains intact in

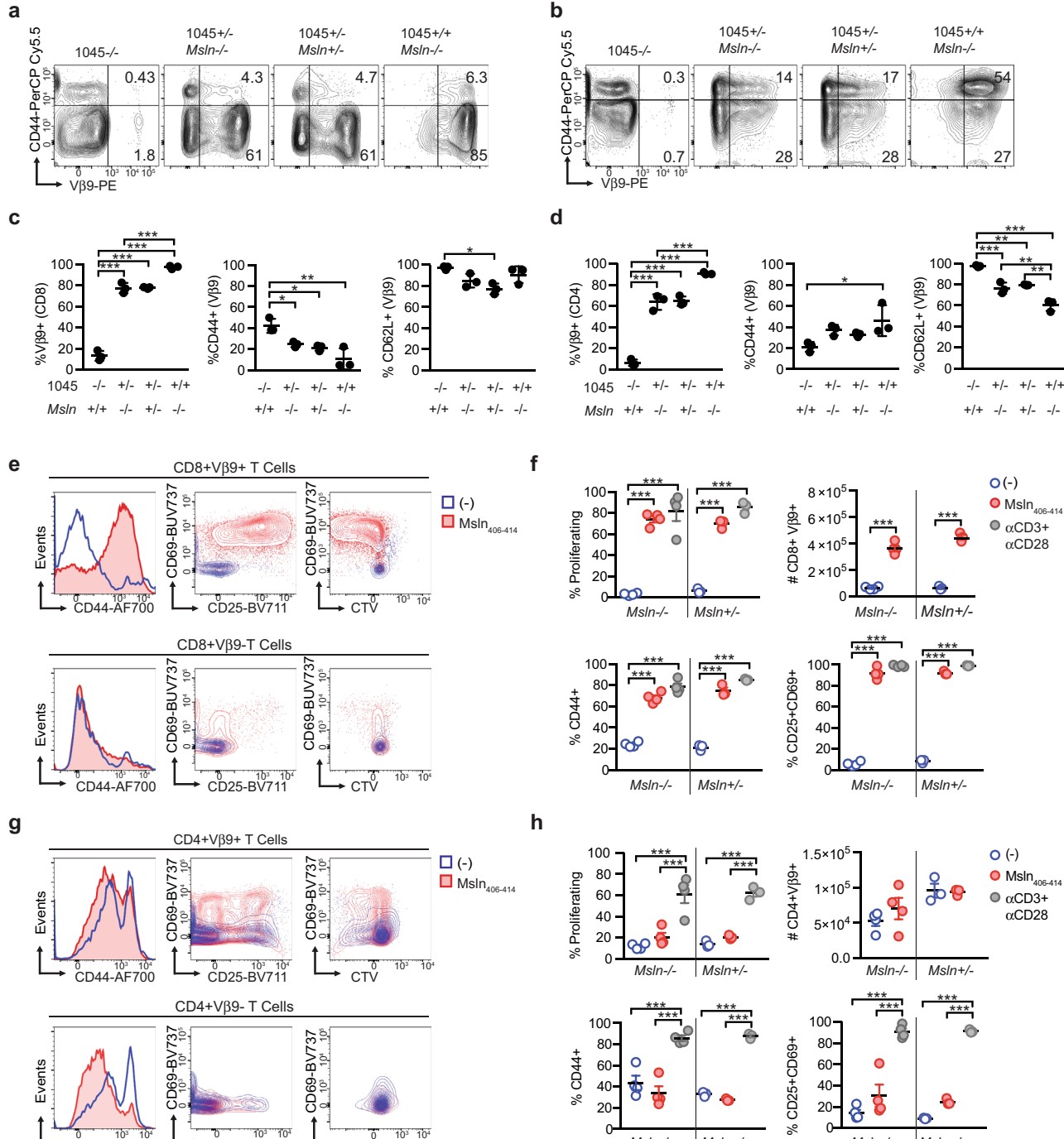

**Fig. 5 | Peripheral 1045 T cells from *Msln*^−/− or *Msln*^+/− mice respond to specific antigen. a** Representative plots gated on splenic CD8 T cells. **b** Representative plots gated on splenic CD4 T cells isolated from the indicated mouse strains. **c** Frequency of splenic CD8 T cells that express Vβ9 (left) and frequency of CD8+ Vβ9+ T cells that express CD44 (middle) or CD62L (right). Data are mean ± SEM. $n = 3$ biologically independent animals. *$p < 0.05$, **$p < 0.005$, ***$p < 0.0005$. One-way ANOVA with a Tukey's posttest. **d** Frequency of splenic CD4 T cells that express Vβ9 (left) and frequency of CD4+ Vβ9+ T cells that express CD44 (middle) or CD62L (right). Data are mean ± SEM. $n = 3$ biologically independent animals.

*$p < 0.05$, **$p < 0.005$, ***$p < 0.0005$. One-way ANOVA with a Tukey's posttest. **e** Representative plots gated on CD8+ Vβ9+ (top row) or CD8+ Vβ9− (bottom row) T cells on day 3 post ± Msln₄₀₆₋₄₁₄ peptide. **f** Quantification of **e** and results post αCD3 + αCD28 on day 3. Data are mean ± SEM. Each dot is an independent mouse. ***$p < 0.0005$. One-way ANOVA with a Tukey's posttest. **g** Representative *p*lots of CD4+ Vβ9+ (top row) or CD4+ Vβ9− T cells (bottom row) on day 3 post ± Msln₄₀₆₋₄₁₄ peptide. $n = 4$ biologically independent experiments. **h** Quantification of **g** and results from post αCD3 + αCD28 on day 3. Data are mean ± SEM. $n = 4$ biologically independent experiments. ***$p < 0.0005$. One-way ANOVA with a Tukey's post test.

TRex mice. However, fewer TRex T cells express endogenous TCRβ than WT T cells and endogenous TCRβ cell surface expression is much lower in TRex T cells vs. WT T cells. Thus, post-transcriptional mechanisms for silencing endogenous TCRβ[47,48] may be playing a role. In the future, endogenous TCRβ could be deleted using CRISPR/Cas9

but this is complicated by the fact that exogenous TCRβ must remain intact. Alternatively, the TRex strategy could be applied directly in a TCRβ-deficient genetic background. An outstanding question, however, is the extent TRex T cells require expression of an endogenous TCRβ for development.

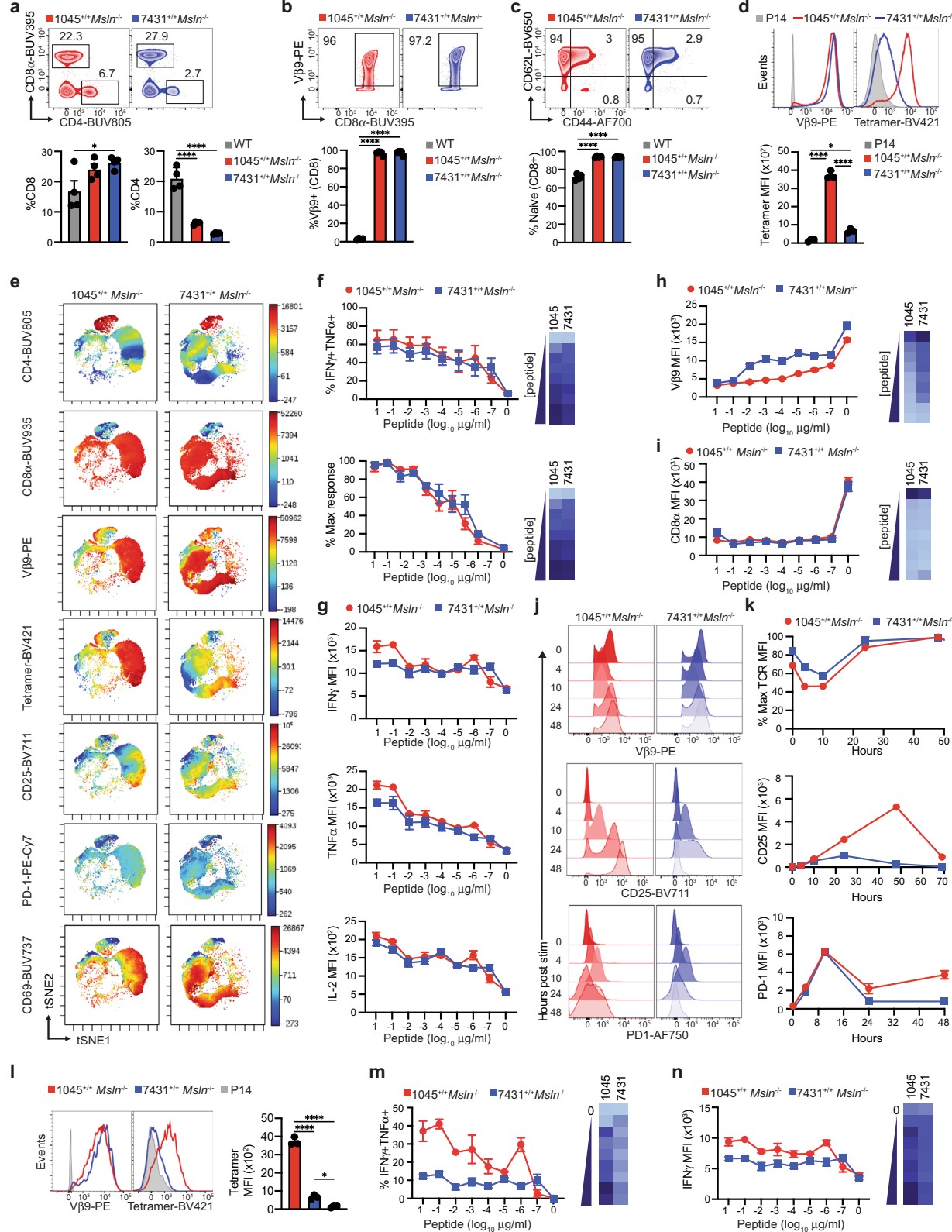

T cells that express high affinity self-reactive TCRs are susceptible to thymic negative selection, an essential central tolerance mechanism that safeguards against autoimmunity. Here, we identify that both copies of *Msln* are necessary for negative selection of high affinity Msln-specific T cells supporting a gene dosage-dependent mechanism of central tolerance. Loss of one *Msln* allele may reduce protein expression on a per cell basis. Alternatively, as *Msln* expression has

been reported in medullary thymic epithelial cells (mTECs)[49] and Aire-dependent genes can be stochastically monoallelically expressed in mTECs[50], *Msln* allele loss may reduce the number of Msln+ thymic APCs capable of mediating negative selection. Fezf2 elicits self-antigen expression in mTECs in an Aire-independent manner[51] and also represses some mTEC genes including *Msln*[52] suggesting *Msln* regulation by Fezf2 may necessitate that both alleles are intact for negative

**Fig. 6 | TCR *Trac* targeting improves the functional avidity of a low affinity TCR.**
**a** Frequency of splenic T cells. Plots (top) and mean ± SEM (bottom). $n = 4$ biologically independent animals. **b** Frequency of splenic Vβ9+ CD8 T cells. Plots (top) and mean ± SEM (bottom). $n = 4$ biologically independent animals. **c** Proportion of splenic Vβ9+ CD8 T cells that express CD44 and/or CD62L. Plots (top) and mean ± SEM (bottom). $n = 4$ biologically independent animals. **d** Vβ9 and $Msln_{406-414}$:H-2D$^b$ tetramer staining gated on CD8 T cells and tetramer MFI. Data are mean ± SEM. $n = 4$ biologically independent animals. **e** ViSNE analysis of T cells day 6 post activation. **f** Frequency of effector CD8 + Vβ9+ T cells co-producing IFNγ and TNFα after antigen restimulation. Normalized maximum (below) and heat map of mean (right). Data are mean ± SEM. $n = 5$ biologically independent animals. **g** Cytokine MFI gated on effector CD8 T cells 5 h post APCs pulsed with titrating $Msln_{406-414}$. Data are mean ± SEM. $n = 5$ biologically independent animals. Vβ9 (**h**) and CD8 (**i**) downregulation 5 h post antigen. Data are mean ± SEM. $n = 2–3$

independent biological replicates. Heat maps of mean (right). **j** Representative histograms of Vβ9, CD25, and PD-1 at the indicated timepoints post incubation with syngeneic splenocytes pulsed with 2 µg/ml of $Msln_{406-414}$. **k** Quantified data from **j**. $n = 3$ biologically independent animals. Data are mean ± SEM. **l** Representative Vβ9 and $Msln_{406-414}$:H-2D$^b$ tetramer staining gated on splenic CD4 T cells (left). Tetramer MFI is graphed as mean ± SEM. $n = 3$ biologically independent animals. **m** Frequency of effector CD4 T cells co-producing IFNγ and TNFα 5 h post incubation with splenocytes pulsed with titrating $Msln_{406-414}$. Heat map of mean (right). Data are mean ± SEM. $n = 2–3$ biologically independent replicates. **n** IFNγ MFI by CD4 T cells following a 5 h incubation with syngeneic splenocytes pulsed with titrating $Msln_{406-414}$ and heat map of mean (right). Data are mean ± SEM. $n = 2–3$ independent biological replicates. For all panels, *$p < 0.05$, **$p < 0.005$, *** $p < 0.0005$, ****$p < 0.0001$. One-way ANOVA with a Tukey's post test.

selection. MSLN is detected in Hassall's corpuscles, which are inflammatory, cornifying, terminally differentiated mTECs[53] in the normal human thymus[9] and single-cell sequencing supports *MSLN* expression in human thymic mesothelial and epithelial cells[54]. *MSLN* is also overexpressed in thymic carcinomas[55]. Thus, further investigation into the thymic cell type(s) that induce negative selection as well as the extent that a TCR affinity threshold exists for central tolerance are warranted.

Prior methods for engineering T cells to express an antigen receptor have primarily used the lentiviral transduction approach. In clinical trials, lentiviral-mediated random CAR integration into *TET2* or *CBLB* caused infused CAR T cell clonal expansion in cancer patients[56,57]. In addition, gene silencing and variable non-uniform receptor expression can occur following retroviral transduction of T cells[22,58,59]. We previously employed γ-retroviral vectors to transduce murine P14 Tg T cells with Msln-specific TCRs[7,60], an approach that necessitated 2 weeks of T cell expansion to obtain sufficient effector T cell numbers for preclinical adoptive T cell therapy studies[7,60]. In contrast to human T cells that can be sustained in vitro over multiple stimulations[61], maintaining murine T cell viability during repetitive in vitro stimulations is more challenging. Further, γ-retroviral vectors can only transduce proliferating cells, precluding the analysis of naïve Msln-specific T cells that could model the activity of MSLN-based cancer vaccines[13,62]. Thus, the 1045 and 7431 *Msln*$^{-/-}$ TRex mice provide a standardized source of native self/tumor-specific T cells. During TCR transduction, exogenous TCRs must also outcompete endogenous TCR chains for CD3 complex and cell surface expression[63]. Exogenous TCR chains mispairing with endogenous TCR chains can result in unknown T cell antigen specificities thereby increasing the potential for toxicity during adoptive cell therapy[60] and may decrease exogenous TCR levels. Indeed, abrogating endogenous TCRβ improved antigen sensitivity in human TCR-transduced T cells[64]. Here, we demonstrate increased antigen sensitivity when the exogenous TCR is located within *Trac* even compared to TCR transgenic T cells with the identical antigen specificity. Many cancers are poorly immunogenic in part through defects in antigen processing and presentation. Thus, targeting an exogenous TCR to *Trac* may be beneficial in such settings. As engineered CD4 T cells contribute to CAR T cell antitumor activity[65] our previous use of P14 TCR transgenic cells as a donor cell source for exogenous TCR transduction limited our assessment to CD8 T cells. We show that the high affinity 1045 TCR has some functions in CD4 T cells from TRex mice permitting future studies to potentiate the antitumor function of MHC I-restricted TCR-engineered CD4 T cells. Thus, Msln-specific TRex mice will permit efficient investigation into how to enhance antitumor T cells in cancer. Particularly, since all T cells express the Msln-specific TCR in TRex mice, experiments to further engineer the T cells to overcome the suppressive tumor microenvironment by expressing chimeric costimulatory proteins, cytokines or performing genetic screens are now readily feasible.

Our study supports that the genomic location of TCR can impact T cell functionality. Despite a lower affinity TCR, 7431 TRex effector T cells were as sensitive to low antigen as 1045 TRex effector T cells. While tetramer staining is not always a surrogate for T cell functionality[66,67], in the past, tetramer staining correlated with increased a higher functional avidity of 1045 RV compared to 7431 RV effector T cells[7], supporting that TCR integration in *Trac* may improve antigen sensitivity of lower affinity TCRs. P14 TRex T cells were also more sensitive to antigen as compared to P14 transgenic T cells, which may be explained in part due to higher TCR expression in P14 TRex T cells. Since pups from multiple independent P14 TRex founders were analyzed, the heightened antigen sensitivity appears reproducible. In addition, we identify that targeting a TCR to *Trac* sustained primary murine T cell function over multiple stimulations in vitro compared to transduced T cells. These data are consistent with a prior study of a CD19 CAR targeted to *TRAC*[22] and exogenous TCR replacement in human T cells[23,43]. Thus, the TRex methodology is an exciting and robust technology to generate heritable alleles encoding a desired TCR and permit investigation into physiological TCR regulation on T cell behavior.

## Methods
### Animals
The University of Minnesota Institutional Animal Care and Use Committee approved all animal studies that conformed to ethical regulations for animal testing and research. Animals were co-housed in SPF vivarium which is maintained at a 14:10 h light:dark cycle, at 68–70°F and 20–70% humidity range. Both female and male mice between the ages of 6–12 weeks old were used in this study. Mice for these studies were euthanized according to IACUC approved methods of $CO_2$ or isoflurane overdose followed by cervical dislocation. C57BL/6J wild type (WT) mice were purchased from Jackson Labs (#000664). Pseudopregnant CD-1 female mice were purchased from Charles River Laboratory (#CD-1 022). Generation of TRex animals was performed in the Mouse Genetic Laboratory at the University of Minnesota. B6.129P2-B2mtm1Unc/DcrJ (*B2m*$^{-/-}$) mice backcrossed to C57Bl/6 strain for 11 generations were purchased from Jackson Labs (#002087) and backcrossed to 1045 TRex mice. P14 and OT1 TCR transgenic were kindly provided by Dr. David Masopust and Dr. Vaiva Vezys (University of Minnesota). P14 TRex and P14 Tg mice were on the *Rag*$^{+/+}$ background. P14 TRex mice were homozygous for the P14 TCR. Three independent P14 TRex founders were bred to generate homozygous F1 pups for the P14 TRex analysis.

### rAAV serotype screening
Splenocytes from WT mice were activated in vitro with 1 µg/ml anti-CD3ε (145-2C11, BD Biosciences) and 1 µg/ml anti-CD28 (37.51, BD Biosciences) in the presence of 10 ng/µl recombinant human IL-2 (Peprotech) and 5 ng/µl recombinant murine IL-7 (R&D Systems) in T cell media (DMEM, 10% FBS, 2 µM L-glutamine, 100 U/ml penicillin/streptomycin, 25 µM 2-β-mercaptoethanol) at 37 °C, 5% $CO_2$. After

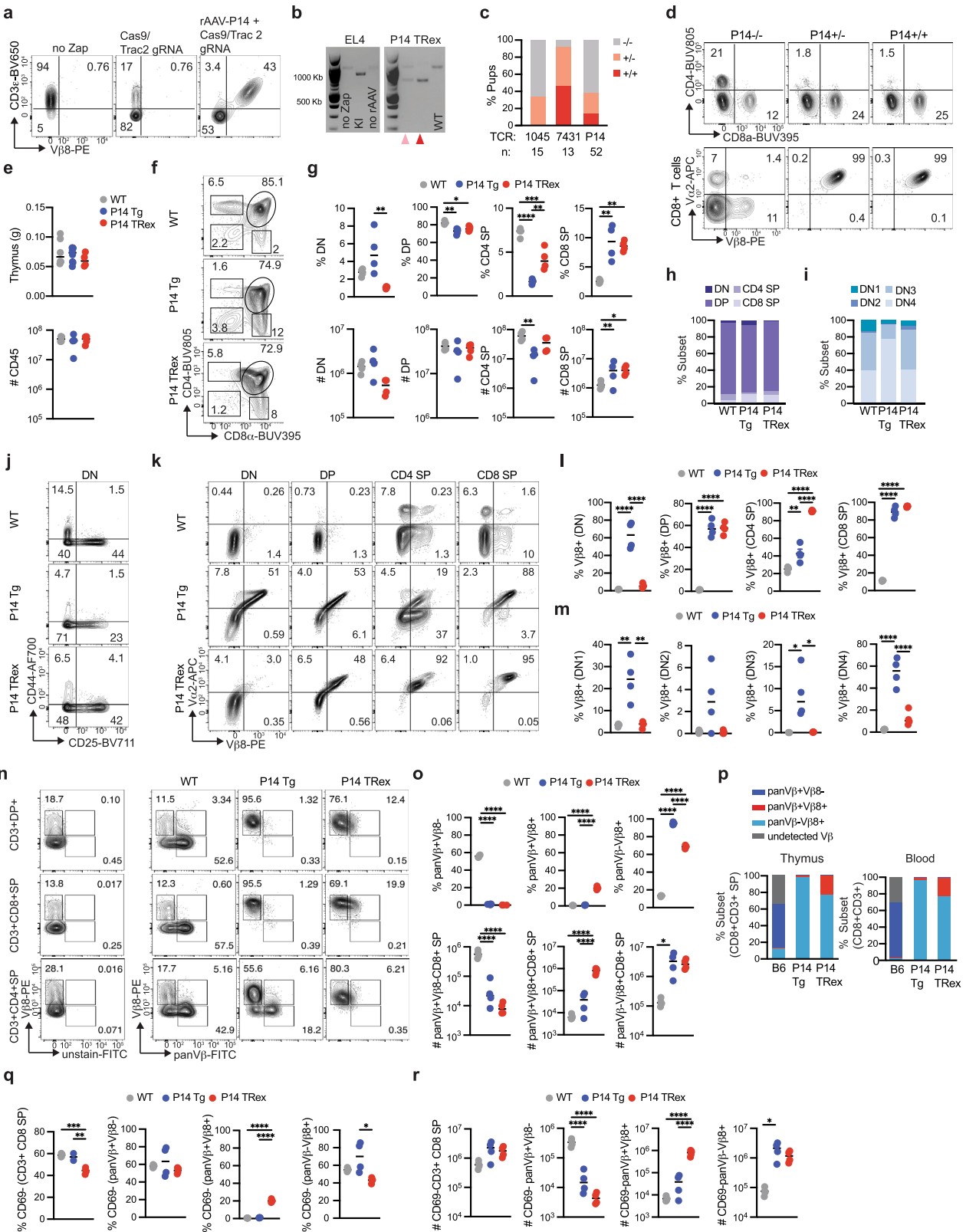

2 days, activated T cells were centrifuged and incubated with rAAV serotypes (UPenn Vector Core) engineered to express GFP. After 3 days, GFP expression in live T cells was determined by flow cytometry.

## Cell lines

EL4 cells are derived from a lymphoma induced in a C57BL/6N mouse by 9,10-dimethyl-1,2-benzanthracene and are commercially available (TIB-93, ATCC). NIH/3T3 fibroblast cell line isolated from a mouse NIH/

Swiss embryo are commercially available (CRL-1658, ATCC). Both cell lines were cultured according to ATCC specifications and were negative for Mycoplasma.

## TCR cloning into pAAV destination vector and virus production

The *Trac* targeting TCR vectors were produced by first designing ~1 kb homology arms (HA) flanking the CRISPR sgRNA target site in exon 1 such that the high affinity (clone 1045[7]) or low affinity (clone 7431[7])

**Fig. 7 | Characterization of T cell development in P14 TRex mice. a** Frequency of EL4 cells that express Vβ8 and CD3ε on day 3 post electroporation with *Trac* sgRNA2 + Cas9 RNP ± rAAV-P14. No zap, negative control. **b** Donor P14 TCR integration was determined by a junction PCR of EL4 DNA (left image) or representative P14 TRex pups (right image). KI, knock-in. Pink arrow, P14$^{+/-}$ TRex pups; red arrow, P14$^{+/+}$ TRex pups. WT, wild-type band. **c** Frequency of TRex pups with the indicated genotype. **d** Frequency of circulating CD4 and CD8 T cells (top, gated on live CD45+ cells) and frequency of Vα2+ Vβ8+ among CD8 T cells from P14 TRex pups. **e** Thymus weight in grams (g) and CD45 cell number per thymus from WT, P14 transgenic (Tg) or P14$^{+/+}$ TRex mice. **f** Plots are gated of CD45+ B220− thymocytes. **g** Frequency (top) and number (bottom, per thymus) of double negative (DN), double positive (DP), CD4 single positive (CD4 SP) and CD8 single positive (CD8 SP) thymocytes. **h** Mean frequency of each subset among total CD45+ B220−. **i** Mean frequency of DN1-DN4 subsets among total DN. **j** DN1-DN4 plots are gated on CD4−CD8− DN thymocytes. CD44+ CD25-(DN1), CD44+ CD25+ (DN2), CD44-CD25+ (DN3), and CD44- CD25- (DN4). **k** Representative plots of Vβ8+ Vα2+ staining gated on the indicated thymocyte subset. **l** Proportion of the indicated subsets that express Vβ8. **m** Proportion of DN subsets that are Vβ8+. **n** Plots of the indicated thymocyte subsets that express exogenous (Vβ8+) and/or endogenous (panVβ+) Vβ. **o** Frequency (top row) and number (bottom row, per thymus) of CD3+ CD8 SP that express exogenous (Vβ8+) and/or endogenous (panVβ+) TCRβ. **p** Mean proportion of CD3+ CD8 SP that express exogenous (Vβ8+) and/or endogenous (panVβ+) TCRβ in thymus or blood. **q** Frequency of mature CD69- CD8 SP thymocytes among total CD3+ CD8 SP and single or dual TCRβ+ CD3+ CD8 SP. **r** Number of CD69- CD3+ CD8 SP per thymus (left) and number of single or dual TCRβ+ CD69-CD3 + CD8 SP thymocytes. For all quantified panels, data are mean± SEM. $n = 4$ mice per group. $*p < 0.05$, $**p < 0.005$, $***p < 0.0005$, $****p < 0.0001$. One-way ANOVA with a Tukey's posttest.

Msln$_{406-414}$:H-2D$^b$-specific or P14 TCR were inserted in-frame. A Furin (RRKR)-GSG-T2A element[68] was incorporated at the 5′ end of the TCR insert site to facilitate co-translational separation from the residual peptide sequence of the endogenous *Trac* locus. The *Trac* HA-TCR-GSG-T2A sequence was synthesized as a gBlock Gene Fragments (IDT, Coralville, IA) with AttB sites and subcloned into pDONR221 using the Gateway BP Clonase II Enzyme Mix (ThermoFisher Scientific, Waltham, MA) to produce pENTR-mTRAC HA. TCR sequences were codon optimized and synthesized by Genscript and subsequently cloned into pENTR-mTRAC HA using Gibson Assembly[69]. Following sequence verification, the pENTR-mTrac HA-TCR was cloned into pAAV-Dest-pA using the Gateway LR Clonase II Enzyme Mix (ThermoFisher Scientific, Waltham, MA). pAAV constructs were sent to Vigene (1045 TCR) or SignaGen Laboratories (7431 and P14 TCR) for commercial AAV production. High titer virus ranged from 1.92–3 × 10$^{13}$ gene copies (GC) per mL and was stored at −80 °C.

## Cas9 ribonucleoprotein and sgRNAs

Synthego sgRNAs were resuspended at 50 μM. 7 μl TrueCut Cas9 v2 (5 μg/mL, ThermoFisher Scientific, A36498) was combined with 7 μl *Trac* sgRNA 2 or 7 μl *Msln* sgRNA 2 at a 1:1 molar ratio and mixed gently by pipetting similar to as described[28]. The *Msln* sgRNA was not used for generation of the P14 TRex mice. Two sgRNAs specific to murine *Trac* exon 1 were tested, including *Trac* sgRNA 1: UCUUUUAACUGGUACACAGC (−54220544) and *Trac* sgRNA 2: UUCUGGGUUCUGGAUGUCUG (−54220521). While both sgRNAs efficiently removed endogenous TCRs, only *Trac* sgRNA 2 resulted in exogenous TCR integration (Fig. 1) and was therefore used in all subsequent experiments. Two sgRNAs specific to murine *Msln* exon 4 were initially tested, *Msln* sgRNA 1: GGAGGUAUCUGACCUGAGCA (−25753010) and *Msln* sgRNA 2 GGCCAAGAAAGAGGCCUGUG (+25753054) and validated in murine B6 3T3 fibroblasts. *Msln* sgRNA 2 was selected for all subsequent experiments.

## Primary murine T cell activation

Spleens were mechanically dissociated using a 40 μm filter. RBCs were lysed by resuspension in 1 ml ACK lysis buffer for 2 min. Lysis was quenched by addition of 10 mls of T cell media. T cells were centrifuged at 350 × *g* for 5 min at 4 °C and resuspended in 10 ml of T cell media containing 10 ng/μl recombinant human IL-2 (rhIL-2, Peprotech), 5 ng/μl recombinant murine IL-7 (rmIL-7, R&D Systems), and 1 μg/ml anti-CD3ε (clone 145-2C11) and 1 μg/ml anti-CD28 (clone 37.51) (BD Biosciences). Alternatively, we used 10 ng/μl recombinant human IL-2 (rhIL-2, Peprotech) and 10 μg/ml Msln$_{406-414}$ peptide (GQKMNAQAI, Genscript) or 10 μg/ml GP33 peptide (KAVYNFATM, Genscript) for studies evaluating antigen-specific T cell activation. Splenocytes were cultured in T25 flask for overnight at 37 °C, 5% CO$_2$. Cells were counted using a hemocytometer and Trypan blue exclusion. A total of 5 × 10$^5$ cells were transferred into a 12 well plates at and incubated at 37 °C, 5% CO$_2$ for 24 h prior to editing, or for longer periods of time as indicated.

## Superovulation and rAAV incubation

24–28-day old female C57BL/6J mice were purchased directly from Jackson Labs (000664) and were superovulated by i.p. injection of 5 IU/mouse of Pregnant Mare Serum Gonadotropin (PMSG, C1063, Sigma). After 47–48 h, 5 IU/mouse of human Chorionic Gonadotropin (hCG, HOR-250, PROSPEC Protein Specialists) was injected i.p. in PMSG-treated females. Superovulating females were immediately crossed with C57BL/6J males at a 1:1 ratio to produce 1-cell zygotes. The next morning, zygotes were collected and washed using standard methods[70]. Briefly, cumulus-oocyte-complex were collected from the ampulla of the plugged females, treated in hyaluronidase (H4272, Sigma) in a 35 mm TC-treated dish (#353001, Falcon) containing 3.5 ml of modified Human Tubal Fluid (mHTF, http://card.medic.kumamoto-u.ac.jp/card/english/sigen/manual/medium/htf.html)[71] for 2 min to remove cumulus cells around the zygotes. The zygotes were then washed 2× in mHTF and then zona pellucida was thinned by briefly treating the zygotes in the Acidic Tyrode's solution (T1788, Sigma). Zygotes were subsequently washed 6X in M2 media (MR-051-F, Millipore), and incubated in 50 μl of mHTF containing rAAV (1.5 × 10$^8$ GC/μl) covered by mineral oil (M8410, Sigma) in a 60 mm tissue culture dish (353004, Falcon) for 6 h at a 37 °C, 5% CO$_2$.

## Electroporation of zygotes with CRISPR Cas9 RNPs

TrueCut Cas9 (ThermoFisher Scientific, A36498) and sgRNAs were combined at a 1:1 molar ratio and incubated for 10 min at room temperature to generate ribonucleoprotein (RNP) complexes and stored on ice during transfer to the University of Minnesota Mouse Genetics Laboratory. Following 6-hour incubation with rAAV, zygotes were washed 1X in Reduced Serum Medium (OPTI-MEM, #31985-062, Gibco). Zygotes were transferred with a pipette and next mixed with 10 μl of OPTI-MEM, 2 μl of rAAV at 1.5 × 10$^9$ GC/μl (final concentration: 1.5 × 10$^8$ GC/μl), and 2 μl of 10X preformed RNP complex (Cas9+gRNAs to *Trac* and *Msln*). The electroporation was performed in a 1 mm gap electroporation cuvette (Cat# 5510, Molecular BioProducts) using the BioRad Gene Pulser Xcell according to the following parameters: square wave at 30 V, 6 pulses with 3 ms duration and 100 ms interval. After the electroporation, zygotes were washed once in 1X OPTI-MEM and then transferred to the original mHTF drop for overnight culture. The next day, 2-cell embryos were transferred into pseudopregnant CD-1 females (Charles River Laboratory) and after 19 days, pups were born. Number of zygotes, embryos, CD-1 females and pups born following zygote engineering using CRISPR/Cas9 and 1045, 7431 or P14 rAAV for generating TRex mice is in Supplementary Table 2.

## PCR genotyping

DNA was isolated from ear snips using the REDExtract Kit (Sigma Aldrich). PCR was run using Q5 HiFi Master Mix (New England Biolabs) for *Trac* KO, *Msln* KO, and *Trac* Junction PCR protocols using the following gene-specific PCR primers purchased from IDT: *Trac* KO forward, 5′-GCTAGATCCTAGGCTGTCATTTC-3′, *Trac* KO reverse, 5′-

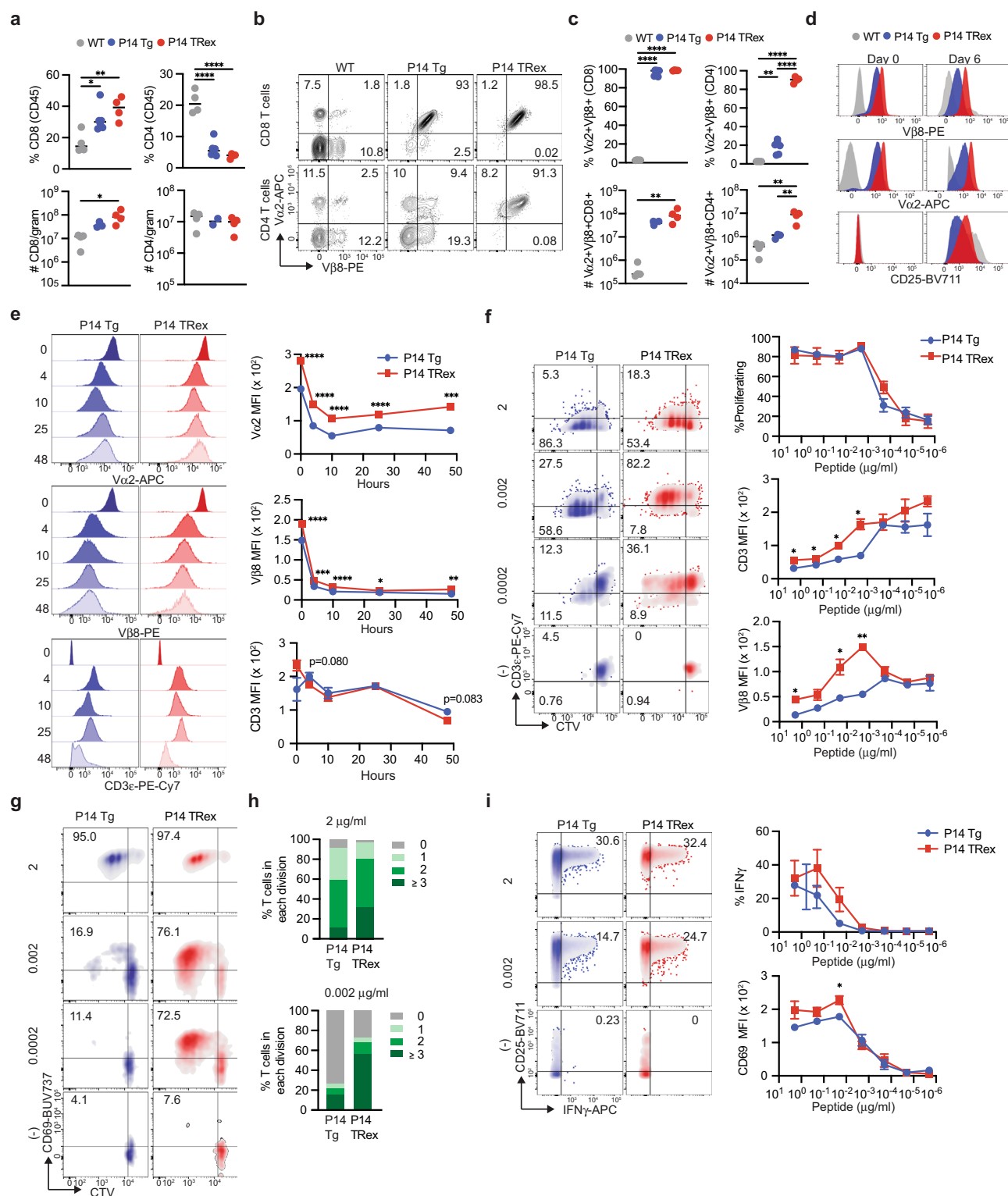

CCAATGTCCTCTGTCATGTTCT-3', with an amplicon length of 579 bp; *Msln* KO forward, 5'-AGGTGGGTTCAGTACCTTTG-3', and *Msln* KO reverse, 5'-GATCAGCTCAGACTTGGGATAG-3', with an amplicon length of 698 bp. Amplification was run for 30 cycles at 95 °C for 30 s, 55 °C for 30 s, 74 °C for 1 min. To assess exogenous TCR integration into the *Trac* locus, we created a *Trac* junction PCR using the following gene-specific PCR primers: wild type (WT) Forward, 5'-CTCTGGTGTGAGTGCTATTC-3', 1045 and 7431 knock-in (KI) Forward, 5'-CCTGTTCTGGTACGTGAGATAC-3', P14 KI Forward, 5'-GTAGCTAT

GAGGATAGCACCTTT-3' and a junction universal reverse primer, 5'-CA AGAGAAGACAGGAAGGTGAG-3'. The WT amplicon length is 1025 bp, the1045/7431 KI amplicon length is 750 bp, and the P14 KI amplicon length is 742 bp. Amplification was run for 30 cycles of 95 °C for 30 s, 60 °C for 30 s, and 74 °C for 1 min. *Trac* and *Msln* KO PCR products were purified using a PCR Clean-Up Kit (Qiagen) and were subsequently submitted for Sanger sequencing through Eurofins genomics using both *Msln* and *Trac* forward and reverse primers described above. PCR was run on an Eppendorf Vapo Protect thermocycler.

**Fig. 8 | Targeting a TCR to *Trac* enhances exogenous TCR expression and antigen sensitivity. a** Frequency (top row) and number (bottom row) of splenic CD4 and CD8 T cells. Each dot is an independent mouse. Data are mean ± S.E.M. $n = 4$ mice per group. *$p < 0.05$, **$p < 0.005$, ****$p < 0.0001$. One-way ANOVA with a Tukey's posttest. **b** Representative plots gated on splenic CD8 (top row) or CD4 (bottom row) T cells. **c** Frequency (top row) or number (bottom row) of Vα2+ Vβ8+ splenic T cells. Each dot is an independent mouse. Data are mean ± SEM. $n = 4$ mice per group. **$p < 0.005$, ****$p < 0.0001$. One-way ANOVA with a Tukey's posttest. **d** Histograms showing the indicated markers ex vivo (day 0) and on day 6 post in vitro activation with gp33 peptide and IL-2. Representative of $n = 3–4$ mice per group. **e** Histograms of TCR chains and CD3 at the indicated hours post activation with 2 μg/ml of gp33 peptide. Data are quantified (right) and mean ± S.E.M. n = 4 mice per group. Significance was determined by unpaired two-tailed T-tests with

Welch correction a false discovery rate (FDR) of 1% and a two-stage step-up[73]. *$p < 0.05$, **$p < 0.005$, ***$p < 0.0005$, ****$p < 0.0001$. **f** Proliferation of CD8 T cells on day 3 post activation with titrating doses of gp33 peptide (y-axis) and rhIL-2 (10 ng/μl). Significance was determined by unpaired two-tailed T-tests with Welch correction with an FDR of 1% and a two-stage step-up[73]. *$p < 0.05$, **$p < 0.005$. **g** Proliferation of CD8 T cells on day 3 post activation with titrating doses of gp33 peptide (y-axis) without exogenous IL-2. **h** Proportion of CD8+ Vβ8+ T cells among each division cycle on day 3 post activation. Data are mean. $n = 4$ mice per group. **i** Proportion CD8 T cells producing IFNγ and expressing CD25 (plots) or CD69 (graphed data on right) on day 3 post activation. Data are mean ± SEM. $n = 4$ mice per group. Significance was determined by unpaired two-tailed T-test with Welch correction with an FDR of 1% and a two-stage step-up[73]. *$p < 0.05$.

Sequence results were analyzed using Snapgene and Interference with Crispr Edits (ICE) software (https://ice.synthego.com). Mutant sequences were directly compared to WT control sequence. *Trac* junction PCR product was run on a 1.5% agarose gel and imaged in a UV transilluminator with ethidium bromide.

### TCR targeting to *Trac* in primary murine T cells and EL4 cells
EL4 lymphoma cells were sorted to >95% CD3+ purity. Primary T cells were activated with anti-CD3 + anti-CD28 as described above and after 2 days, T cells were centrifuged for 10 min at 200 × g at 4 °C. Primary T cells and EL4 cells were resuspended at $1 × 10^6 – 1 × 10^7$ cells per ml in P4 solution with supplement (Lonza, V4XP-4024). Synthego sgRNAs were resuspended at 50 μM. 10X RNPs were generated by mixing Synthego sgRNAs and TrueCut Cas9 Protein v2 (ThermoFisher Scientific, A36498) at a 1:1 molar ratio and incubating at room temperature for 10 min. RNPs were diluted tenfold in the cell suspension and cells were transferred to the nucleofection cuvette and incubated at room temperature for 2 min with the cover on. Using the Amaxa 4D Nucleofector, cells were pulsed with pulse code CM137 and allowed to rest 15 min in the cuvette. Cells were diluted 1:10 in prewarmed T cell recovery media (T cell media with no antibiotics) in the cuvette and allowed to recover at 37 °C for 15 min. T cells were transferred to pre-warmed (37 °C) T cell media containing rhIL-2 (10 ng/μl), rmIL-7 (5 ng/μl) and various virus concentrations of rAAV6 engineered to express the 1045 TCR (Vigene), 7431 TCR (Signagen), or P14 TCR (Signagen) homology donor DNA for a total of 30 min after nucleofection. T cells were returned to the incubator (37 °C, 5% $CO_2$) for an additional 3 days prior to flow cytometry and/or DNA sequencing analysis. Typically, EL4 and primary T cells were ~50% viable following this protocol.

### Preparation of mononuclear cells from spleen and blood
Spleens were mechanically dissociated to single cells. RBCs were lysed by incubation in 1 mL of Tris-ammonium chloride (ACK) lysis buffer (GIBCO) for 1–2 min at room temperature. 9 mL of T cell media was added to quench lysis. Cells were spun at 1400 rpm for 5 min at 4 °C and resuspended in T cell media and stored on ice until further use. For PBMCs, 100–200 μl of blood was collected per animal in 20 mM EDTA in a 96-well round bottom plate. RBCs were lysed by resuspension in 150 μl ACK lysis buffer (GIBCO) for 10 min at room temperature. 1 mL of T cell media was added to quench cell lysis. Cells were spun at 350 × g for 5 min at 4 °C, supernatant decanted. Cells were stored in T cell media on ice until further use.

### Monoclonal antibody staining for flow cytometry
Mononuclear cells were stained with various antibodies or $Msln_{406-414}$:H-2D$^b$-APC or -BV421 tetramer (1:100) in the presence of Fc block (αCD16/32) and a live/dead stain (Tonbo Ghost dye in BV510 or APC ef780). Antibodies were diluted 1:100 in FACs Buffer (2.5% FBS + PBS + 1% NaN₃) unless otherwise indicated. All antibodies used for flow cytometry and T cell activation are described in Supplementary Table 3. For intracellular Foxp3 and Ki67 staining, cells were fixed using

Foxp3 transcription factor reagent (Tonbo) for 30 min at 4 °C, washed and intracellular stained with diluted antibodies in permabealization buffer (Tonbo) and stained overnight. The next day, cells were washed 2X with perm/wash buffer and resuspended in FACs buffer. Cells not stained intracellularly were fixed in 0.4% PFA for 15 min at 4 °C, washed 2X with FACS buffer and resuspended in FACs buffer. For dual TCRβ detection, thymocytes or PBMCs were stained with a panel of non-FITC antibodies including Vβ8 for 45 min at 4 °C. Cells were washed 2X with FACS buffer and resuspended in a pool of anti-mouse TCR Vβ-FITC antibodies (10 μl per reaction for each antibody, Supplementary Table 3). Note that Vβ8.1/2 and Vβ8.3-FITC antibodies were excluded from the pooled TCR Vβ master mix. All samples were resuspended in FACs buffer and 100 μl of Countbright Absolute Counting Beads (Thermo Fisher). Cells were acquired with a Fortessa 1770 or Fortessa X-20 using Facs Diva software (BD Biosciences). Data were analyzed using FlowJo software (version 10).

### Intracellular cytokine staining
Splenic mononuclear cells were activated in vitro with Msln or gp33 peptide (10 g/ml) or anti-CD3 + anti-CD28 as described above with rhIL-2 (10 ng/μl, Peprotech). On day 6 post activation, $1 × 10^5$ effector T cells were centrifuged and resuspended with congenic (CD45.1+) pulsed with titrating concentrations of peptide-pulsed splenocytes at a 5:1 APC to T cell ratio. Cells were incubated in round-bottom 96-well plates in a total volume of 200 μl of T cell media + Golgiplug + Golgistop (BD Biosciences) for 5 h at 37 °C, 5% $CO_2$. Cells were centrifuged, and resuspended in cell surface antibodies including CD45.1 to exclude APCs and additional antibodies (Supplementary Table 3) diluted in FACs buffer and incubated for 30 min in the dark at 4 °C. Cells were washed 2X with FACs buffer, fixed and permeabilized (BD Biosciences Fixation Kit) and subsequently incubated with antibodies specific to IFNγ and TNFα diluted in permeabilization buffer overnight in the dark at 4 °C. Cells were washed 2X and resuspended in FACs buffer and 100 μl of Countbright Absolute Counting Beads (Thermo Fisher). Acquisition was performed using a Fortessa 1770 or Fortessa X-20 using FACs Diva software (BD Biosciences). Data were analyzed using FlowJo software (version 10).

### Cell proliferation assay
Live mononuclear splenocytes were counted using a hemocytometer and trypan blue exclusion. $2 × 10^6$ splenocytes were incubated with 5 μM Cell Trace™ Violet (CTV) (Invitrogen) diluted in PBS and incubated for 20 min in the dark at 37 °C, 5% $CO_2$. Cells were washed 4X with T cell media to remove excess CTV and $7.5 × 10^5$ CTV labeled splenocytes were plated in duplicate in 96-well round bottom plates in T cell media with 10-fold serial dilutions of gp33 or Msln peptide ± 10 ng/μl of rhIL-2. Cells were incubated in the dark for 3 days at 37 °C, 5% $CO_2$, stained for various cell surface markers and analyzed by flow cytometry. In Fig. 8i, duplicate plates were also set up in which Golgiplug + Golgistop (BD Biosciences) were added for 5 h prior to cell surface staining, washing, fixing and intracellular staining for IFNγ as

above. Samples were acquired in the Center for Immunology on a Fortessa 1770 or Fortessa X-20 using FACs Diva software (BD Biosciences). Data were analyzed using FlowJo software (version 10).

## ViSNE analysis

ViSNE analysis was performed by gating on total live T cells with default settings of 1000 iterations, 30 perplexity and theta of 0.5 using Cytobank software.

## Msln$_{406-414}$:H-2D$^b$ tetramer production

H-2D$^b$-restricted biotinylated monomer was produced by incubating Msln$_{406-414}$ peptide with purified H-2D$^b$ and β2 m followed by purification via Fast Protein Liquid Chromatography system (Aktaprime plus, GE health care) similar to as described[72]. Biotinylated monomer was conjugated to streptavidin R-APC or R-BV421 (Invitrogen) to produce fluorescent Msln$_{406-414}$/H-2D$^b$ tetramer.

## Cell numbers normalized to tissue gram

The number of live CD45+ cells collected per tissue was determined by FlowJo analysis software and the equation: # CD45+ cells per tube $(n)$ = (# Beads/# Cells) x (Concentration of beads x Volume of beads added). Total number of cells collected from the entire single-cell suspension was determined by multiplying n by total number of stains. Cell numbers were normalized to tissue weight or organ as indicated.

## Immunofluorescence

Tissues were embedded in OCT (Tissue-Tek) and stored at −80 °C. 7 μm sections were cut using a Cryostat and fixed in acetone at −20 °C for 10 min. Sections were rehydrated with PBS + 1% bovine serum albumin (BSA) and incubated for 1 h with primary antibodies to rat anti-mouse Msln (MBL, B35, 1:100) diluted in PBS + 1% BSA at rt. Slides were washed 3× in PBS + 1% BSA and incubated with anti-rat AF546 (Invitrogen, 1:500) for 1 h in the dark at room temperature. Stained slides were then washed 3× with PBS + 1% BSA, washed 3× with PBS, and mounted in DAPI Prolong Gold (Life Technologies). Images were acquired on a Leica DM6000 epifluorescent microscope at the University of Minnesota Center for Immunology using Imaris 9.1.0 (Bitplane).

## Statistical analysis

For power of 80%, the level of significance was set at 5%, 3–6 mice in each group were estimated and no data were excluded. Data are a minimum of two independent experiments. Data were compiled using Microsoft Excel (v16.15.1) and all statistical analyses were conducted using Prism (version 9.0). Appropriate statistical methods were used to calculate significance as described in the figure legends. Unpaired, two-tailed student's $T$ test was used to compare 2-group data unless otherwise indicated. One-way analysis of variance (ANOVA) and Tukey post-test were used for >2-group data. In Fig. 8 panels e, f, and i, significance was determined by an unpaired T-test with a Welch correction and a false discovery rate (FDR) of 1% and a two-stage step-up[73]. Graphed data are presented as mean ± standard error of the mean (SEM) unless otherwise indicated and $p < 0.05$ was considered significant. *$p < 0.05$, **$p < 0.005$, ***$p < 0.0005$, and ****$p < 0.0001$.

## Reporting summary

Further information on research design is available in the Nature Portfolio Reporting Summary linked to this article.

## Data availability

Source data are provided as a Source Data file. Data generated in this study are included in this published article (and its supplementary information files). Source data are provided with this paper.

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

## Acknowledgements

We thank Kris Hogquist and Maude Ashby for helpful discussions on thymocyte development. We thank Sathi Wijeyesinghe for the providing the P14 TCR DNA sequence. We thank Andrew Soerens, Ph.D. for assistance in generation of the Msln$_{406-414}$ monomer. We thank Zoe Schmiechen for providing expertise for ViSNE analysis and Iris Wang and Bryan Sanchez for assistance with animal husbandry. We acknowledge the University of Minnesota Mouse Genetics Laboratory, Flow Cytometry Core, Research Animal Resource (RAR), and the Center for Immunology (CFI) imaging core. A CFI New Mouse Award (to I.M.S.), U of M Brainstorm award (to I.M.S. and B.R.W.) and NIH R03-AI144840 (to B.R.W. and B.S.M.) provided financial support. M.R.R. is supported by National Institutes of Health (NIH) T32 AI 007313 and the U of M Dennis Watson Fellowship. E.J.S. is supported by NIH T35 AI118620 and the AOA Carolyn L. Kuckein Student Research Fellowship. A.L.B. is supported by a computational training award from the American Association of Immunologists. I.M.S. is supported by NIH R01 CA249393, R01 CA255039, NIH P01 CA254849, DOD W81XWH2110525, AACR Pancreatic Cancer Action Network Career Development Award (17-20-25-STRO), an AACR Pancreatic Cancer Action Network Catalyst Award (19-35-STRO), an American Cancer Society RSG RSG-21-102-01-IBC, and pilot awards from the Masonic Cancer Center and Cancer Research Training Initiative (University of Minnesota Medical School).

## Author contributions

B.S.M., B.R.W., and I.M.S. conceptualized the study. M.R.R. executed most of the experiments. B.R.W. designed rAAV TCR vectors and assisted with zygote engineering protocol. W.S.L. assisted with TCR vector generation. J.F.R. developed the TCR *Trac* knock-in protocol in murine primary T cells and the junction PCR. E.J.S. performed the Msln immunofluorescent staining of tissues. E.A.M. managed the maintenance and genotyping of the TRex colonies. E.A.M. and J.Z.B. performed experiments. A.L.B. assisted with experiments. Y.Y. performed zygote engineering. M.R.R. and I.M.S. analyzed and interpreted the data and generated the figures. M.R.R. and I.M.S. wrote the manuscript. M.R.R., B.R.W., and I.M.S. edited the final version of the manuscript. B.S.M., B.R.W., and I.M.S. secured funding. B.R.W. and I.M.S. supervised the study.

## Competing interests

Invention disclosures have been filed with the University of Minnesota related to the generation and use of TRex mice by B.S.W., B.R.W., and I.M.S. The remaining authors declare no competing interests.
