## [Peer Review File · Nature Communications]

Germline T cell receptor exchange results in physiological T cell development and functionREVIEWER COMMENTS

Reviewer #1 (Remarks to the Author):

The authors demonstrate that they can quickly produce TCR transgenic mice with physiological expression (at least of TCR α) due to integration by Knock-in of the TCR β into the Trac locus using a CRISPR approach. They can also knock-out the relevant antigen at the same time, in order to avoid tolerance. This is potentially a useful addition to the immunological toolbox. However, there are a number of important caveats that would need to be resolved before publication.

There is no analysis of the thymus. Since the TRex mice have both TCR alpha and beta knocked-into the Trac locus, under Trac control, there should be no expression until the DP thymocyte stage, after beta-selection. This means that there should be normal beta chain rearrangement and expression prior to the transgenes' expression. Thus, no allelic exclusion of endogenous beta chains. Allelic exclusion is not mentioned once in the paper. We need to see Vb9 expression as well as other Vb expression and CD3 expression in relation to CD4 and CD8 expression in the thymus. Not only percentages, but absolute numbers, need to be shown. CD69 expression in relation to CD3 and Vb9 (and other Vb) expression, should be performed, in order to see effects on positive selection signaling. In fact, the authors say in the Discussion that "One limitation of our study is that the endogenous TCR β chain remains intact in TRex mice, and thus we have not ruled out the possibility that endogenous TCR β chains are co-expressed in TRex mice." This is a really important point that will color all the rest of the analysis. It is crucial that this is investigated.

The finding that 1045 T cells are not tolerized in Msln $^{+/+}$ animals can be explained by their getting through thymic negative selection using a different beta chain.

"Both 7431 and 1045 T cells exhibited a broader spectrum of cell surface TCR as compared to P14 T cells (Fig. 5B)." They also express a wider spectrum of CD8 expression. This is not mentioned. Fig. 5I only shows Vb9 downregulation. What about CD3 expression? This would speak to my query about other beta chains being expressed. The data show that the "low affinity" TCR is more responsive to low amount of antigen than is the "high affinity" TCR. The idea that increased downregulation of the high affinity TCR leads to lower cytokine production in response to low amounts of antigen doesn't really make sense to this reviewer.

"Unexpectedly, antigen encounter induced P14 TCR downregulation to a greater extent as compared to both 7431 and 1045 T cells (Fig. 5K-L)." This could be explained by a proportion of the TRex cells having endogenous beta, whereas P14 shows allelic exclusion.

"Over 90% of CD4 T cells expressed the Msln specific TCR in 1045 and 7431 mice (Fig. 6A). In contrast, only 30-40% of CD4 T cells expressed the gp33-specific TCR in P14 mice (Fig. 6A)." Unfortunately, this is meaningless since they define the Msln-specific TCRs based on Vb9 expression, but the P14 TCR based on Va2 expression (they could have used Vb8). The result for CD4 cells in P14 is due to rearrangement of endogenous Va-regions, impossible in the TRex mice. However, I suspect they would get a result similar to that for the Msln-specific TCRs if they tested for anti-Vb8 in the P14 cells. It is still possible in my view that there are plenty of other Vb elements expressed in the Msln-specific T cells.

"While CD4 T cells isolated from 7431 and 1045 KI mice expressed similar TCR based on Vb9 staining (Fig. 5L), CD4 T cells from 1045 mice stained brighter for Msln tetramer compared to CD4 T cells from 7431 mice (Fig. 5L)." Fig 5L and K are misnamed. Really parts of Fig 6.

The finding that "...that Tregs accumulate during in vitro expansion of P14 T cells (with RV expression of Msln-specific TCRs) (Fig. 2G-H)." can be explained by the P14 cells having no endogenous b-chains but also no allelic exclusion of endogenous alpha chains, so a lot of possibilities for different pairings. The results of Fig 6F,G (not 5 as stated) in fact imply that the P14 cells that turn into Tregs express non-Va2 alpha chains, and it is entirely possible that the Msln-specific Tregs can express other Vb than Vb9 – this really should be checked. Also, the staining of P14 with aVb8, not just aVa2, should be tested.

Minor points:

P5. "...as determined Va2 staining" should be "as determined by Va2 staining"

P9. Needs editing: "CD4+Vb9+ T cells from 1045 TRex mice die upregulated CD69 following Msln406-414:H-2Db recognition..."

P10. "... (Fig. 5G), which was inconsistent tetramer staining." Should read "inconsistent with tetramer"

Reviewer #3 (Remarks to the Author):

The MS by Stromnes et al describes the generation of mice with a quasi monoclonal TCR repertoire through CRISPR mediated knockin of rearranged TCRA and TCRB chains from two different Msln specific TCRs. The authors discuss that this strategy i) represents an accelerated approach to the generation of TCR transgenic mice as compared to 'classical' approaches and ii) is a way to achieve more physiological expression of TCRs (as compared to conventional TCR transgenics) with 'improved fidelity'.

Point i) is thoroughly addressed in the first half of the MS and the technology and methods that were applied seem well described, also in comparison to introduction of the very same TCRs into mature T cells through retroviral transduction. This is interesting, and to my knowledge the first description of the heritable and orthotopic introduction of TCRs into the mouse germline using this approach. A major achievement.

Point ii) deals with the question of whether the gene replacement strategy results in T cells with 'more physiological' features (vis-a-vis conventional TCR transgenics) regarding timing of TCR expression, selection into distinct T cell lineages and responsiveness to cognate antigen. Unfortunately, the authors in this context compare features of the newly generated Trex systems with existing TCR transgenic systems of an unrelated TCR specificity (mostly the P14 TCR, and in a few instances the OTI system). However, it is very questionable whether these results justify the statement that TCR exchange results in 'higher fidelity' of the emerging T cells. First, it remains somewhat unclear what 'fidelity' is. Second, and way more importantly, using Trex cells and conventional transgenic cells with different TCRs is not suited to elucidate and compare fundamental physiological features (selection, response to antigen) of cells that have been generated in one or the other way. The only way to systematically do this would be to work with a single, well-defined TCR, for instance to 'recreate' the P14 TCR as Trex system. Otherwise, it will not be possible to make any conclusive statements regarding the question whether differences between conventional TCR transgenic cells and Trex cells may actually relate to the methodology that was used to generate these systems or may instead merely reflect inherent features of two distinct TCRs.

In sum, the 'technological' part of the manuscript presents exciting data, certainly of substantial interest to the field. I think a careful analysis of T cell selection in the Trex models (perhaps in conjunction with an extended 'biological' focus on a comparison to Trex mice in which Msln is concomitantly knocked out) would have been sufficient for an exciting MS. Unfortunately, in its present form, the 'biological' part of the MS (and also the title) presents conclusions that cannot be drawn from comparing unrelated TCR specificities and in my opinion is therefore not suitable for publication.

Additional point:

The MS appears to be written in a very sloppy manner. For instance, already the summary is very difficult to understand. There is an excessive number of typos, and on multiple occasions, numbering of display items in the text and in the figures does not correspond to each other, in particular also in figure legends. Too many to list, and it is not the reviewer's task to proofread a MS. This made reviewing the MS very cumbersome.

We sincerely thank the reviewers for their time, effort, and insightful feedback. We substantially revised the manuscript including generating and characterizing a new mouse strain (P14 TRex, new Figures 7 and 8) and analyzing thymocyte development in two TRex models (see new Figures 4 and 7). We substantially revised the text of the manuscript including the abstract, introduction, results, discussion, figure legends, supplementary figures, and methods to account for our new findings and to address the critique. Jonah Butler has been added as an author as he performed experiments for the revision. In short, we believe our manuscript is much improved and that our interpretations are substantiated by the data presented. We anticipate that the novel methodology to generate physiological antigen specific T cells as well as the strains reported herein will be broadly impactful for the immunology field. We are grateful for the opportunity to revise our manuscript and believe it is now well-suited for publication in *Nature Communications*. All major changes in the manuscript are in blue. Please see below the point-by-point response to reviewer comments:

REVIEWER COMMENTS

Reviewer #1 (Remarks to the Author):

The authors demonstrate that they can quickly produce TCR transgenic mice with physiological expression (at least of TCR α) due to integration by Knock-in of the TCR α b into the Trac locus using a CRISPR approach. They can also knock-out the relevant antigen at the same time, in order to avoid tolerance. This is potentially a useful addition to the immunological toolbox. However, there are a number of important caveats that would need to be resolved before publication.

Thank you for your positive response to our manuscript.

1. There is no analysis of the thymus. Since the TRex mice have both TCR alpha and beta knocked-into the Trac locus, under Trac control, there should be no expression until the DP thymocyte stage, after beta-selection. This means that there should be normal beta chain rearrangement and expression prior to the transgenes' expression. Thus, no allelic exclusion of endogenous beta chains. Allelic exclusion is not mentioned once in the paper. We need to see Vb9 expression as well as other Vb expression and CD3 expression in relation to CD4 and CD8 expression in the thymus. Not only percentages, but absolute numbers, need to be shown. CD69 expression in relation to CD3 and Vb9 (and other Vb) expression, should be performed, in order to see effects on positive selection signaling. In fact, the authors say in the Discussion that "One limitation of our study is that the endogenous TCRb chain remains intact in TRex mice, and thus we have not ruled out the possibility that endogenous TCRb chains are co-expressed in TRex mice." This is a really important point that will color all the rest of the analysis. It is crucial that this is investigated.

We agree with the reviewer that the lack of thymus analysis was a major weakness in the original submission. We have tried our best to fully address this concern in the revision as follows:

- We added two new main figures (Figure 4 and Figure 7) and two supplemental figures (Supplementary Figure 4 and 7) that investigate thymocyte development in two independently derived TRex mouse models.
- We show 1045 $^{+/+}$ TRex thymocyte maturation in Msln $^{+/+}$, Msln $^{+/-}$ and Msln $^{-/-}$ backgrounds while also including wild type (WT) C57Bl6/J mice as a control. Percentages and numbers of thymocytes in sequential stages of development (DN, DP, SP) are provided as well as the stage at which the exogenous 1045 TCR is expressed (see new Fig. 4 and Sup. Fig. 4). These new results are discussed in the results on pages 7-8.

- We show that 1045 T cells undergo negative selection when both copies of *Msln* are present (Fig. 4b-d). This new result suggests that tolerance to *Msln* is gene dosage dependent and we discuss this in more detail in the revised discussion (see page 16).
- We show that the 1045 TCR can be first detected in DN, specifically DN4, which are likely transitioning to DP (new Fig. 4e,f,h) and consistent with timing of initial opening of the *Trac* locus.
- We show that MHC class I is required for positive selection of the 1045 TRex TCR expressing CD8 T cells in the thymus (new Fig. 4j-l).
- Based on reviewer #3 comments below, we rapidly generate a new P14 TRex strain using the same CRISPR-READI approach (see Sup. Table 3). We compare endogenous $V\beta$ in P14 TRex vs. P14 transgenic and WT mice (see new Fig. 7 and Sup Fig. 7). We find that ~20% of the P14 TRex thymocytes and circulating T cells co-express an endogenous $V\beta$. Further, all of the T cells that express an endogenous $V\beta$ also express the exogenous P14 $V\beta 8$, which is significantly less than the frequency of WT cells that express an endogenous $V\beta$ (see new Figure 7n-p). Further, endogenous $V\beta$ levels, as measured by mean fluorescence intensity, are significantly lower in TRex T cells as compared to WT T cells (see new Sup Fig. 7e). We also assess endogenous $V\beta$ in the 1045 TRex strain in the periphery. In this strain, about 35-40% of 1045 TRex T cells co-express an endogenous $V\beta$ with the exogenous $V\beta 9$ (Sup Fig. 7j). Thus, the data supports that there is a lack of allelic exclusion at the beta chain in some of the T cells. However, the data also suggest endogenous $V\beta$ may be undergoing post-transcriptional silencing in some of the TRex T cells. These new data are described in the results (pages 11-12) and discussion (page 15-16) of the revision.
- We performed a detailed thymocyte analysis of P14 TRex vs. P14 transgenic T cells that support that TRex T cells undergo normal physiological stages of thymocyte maturation (new Figure 7, and Sup Fig. 7). Similar to 1045 TRex model (see new Fig. 4 and Sup. Fig. 4), the exogenous P14 TCR can be first detected in DN4 in TRex mice (new Figure 7m), which contrasts with P14 transgenic T cells in which the transgenic TCR is first detected in DN1 (new Figure 7l-m).
- As requested, we analyzed the expression pattern of CD3 in the thymus in relation to CD4 and CD8 and then further analyzed the expression of CD69. We show that CD8+CD3+ SP can be further distinguished by CD69 and CD62L into immature (CD69+CD62L-) and mature (CD69-CD62L+) subsets in TRex mice (see new Supplementary Figure 7l for representative staining). We show the frequency and number of CD3+DP, CD3+CD4 SP and CD3+CD8 SP that express $V\beta 8$ alone or with an endogenous $V\beta$ in P14 TRex vs. P14 Tg and WT mice (new Figure 7n-p). We show the proportion of each subset (dual $V\beta$ expressing vs. single $V\beta$ expressing) that lack CD69, as a marker of CD8 SP maturation (see new Figure 7q-r). These new data are described in the results on pages 12-13.

2. The finding that 1045 T cells are not tolerized in *Msln*^{+/+} animals can be explained by their getting through thymic negative selection using a different beta chain.

In the original submission (old Figure 4, which is now Figure 5 in the revision and is unchanged from the original submission) we show equal percentages of CD8+ $V\beta 9$ ⁺ cells in the spleen of *Msln*^{-/-}, *Msln*^{+/-} and *Msln*^{+/+} mice (Figure 5A of current manuscript) indicating that not all T cells are undergoing central tolerance in *Msln*^{+/+} mice. Of note, we compared proliferation between 1045 *Msln*^{-/-} and *Msln*^{+/-} T cells and found no difference in T cell activation markers or proliferation. However, the direct comparison to peripheral T cells from *Msln*^{+/+} was not included in the original or new submission. In the revised manuscript, as pointed out above, we provide new data that 1045 T cells appear to undergo negative selection in the thymus only when both copies of *Msln* are expressed (see new Fig.

4b-d). Thus, there may be a tolerant phenotype in peripheral T cells when both copies of Msln are expressed. To fully uncover the extent that there is peripheral tolerance, and the precise mechanism of tolerance, is beyond the scope of the current manuscript but is a project the lab is currently pursuing.

3. “Both 7431 and 1045 T cells exhibited a broader spectrum of cell surface TCR as compared to P14 T cells (Fig. 5B).” They also express a wider spectrum of CD8 expression. This is not mentioned. We appreciate the reviewer bringing this specific point to our attention. Due to comments from Reviewer #3 (see point #2 below), as well as advice from the Editor, our original comparison of 7431 and 1045 TRex T cells to P14 transgenic T cells was not ideal because TCR specificities were different. Therefore, in the revised manuscript, we no longer compare the 1045 and 7431 TRex T cells to P14 transgenic T cells (see new Figure 6) and thus the above statement is removed from the text. We revised Figure 6 to remove the P14 transgenic T cell data from all panels, except in Figure 6d where it serves as a negative control for Msln tetramer staining.

As mentioned above, in the revision, we now describe the generation and characterization of a new P14 TRex strain, using the same CRISPR-READI approach, and include a new and extensive analysis of thymocyte development (new Figure 7) and peripheral T cell function (new Figure 8) in these P14 TRex mice compared to P14 historical TCR transgenic. This removes the variable of different TCR specificities and thus we can compare directly the TRex approach to historical TCR transgenics. From these new data, we conclude that TCR, as measured by V β 8, V α 2, and CD3e is higher in P14 TRex T cells vs. P14 transgenic T cells at baseline and following activation (see new Figures 8d-e, new Supplementary Figure 8d). Further, while CD8 mean fluorescence intensity (MFI) was similar in P14 Tg vs. P14 TRex at baseline (e.g., ex vivo analysis of CD8⁺ splenic T cells) we did note a slight increase in CD8 MFI in P14 TRex T cells as compared to P14 Tg T cells 6 days after activation (see new Supplementary Figure 8d). CD8 MFI was similar in P14 TRex T cells as compared to polyclonal wild type CD8 T cells from syngeneic C57Bl6/J mice (see new Sup. Figure 8d). Thus, we conclude that targeting a TCR to *Trac* may impact TCR cell surface levels and can lead to a modest effect on CD8 coreceptor expression. We state this in the text as follows, “*TRex T cells expressed more CD3e, V α 2 and V β 8 ex vivo (day 0) and following activation (day 6) than analogous P14 Tg T cells (Fig. 8d, Supplementary Fig. 8d). CD8 MFI was comparable ex vivo and modestly higher after activation in TRex T cells (Supplementary Fig. 8d).*” (page 13)

4. Fig. 5l only shows Vb9 downregulation. What about CD3 expression? This would speak to my query about other beta chains being expressed.

Please see response to point #1 above regarding how we determined endogenous V β chain expression in the TRex model. Please note that Figure 5l in the old manuscript is now Figure 6l in the revised manuscript and that the P14 comparison to 1045 and 7431 TRex mice has been removed from new Figure 6 (see response to point #3 above). To address TCR and CD3 downregulation, we compared CD3 and TCR downregulation in P14 TRex vs. P14 Tg mice (see new Fig. 8e-f). During primary activation, TCR and CD3 downregulation kinetics are similar in TRex vs. Tg mice in response to 2 μ g/ml of peptide (Figure 8e). However, we do find that both TCR and CD3 downregulation are less pronounced in TRex vs. Tg effector T cells when antigen is limiting (see new Figure 8f). We describe these data on pages 13-14 in the results.

5. The data show that the “low affinity” TCR is more responsive to low amount of antigen than is the “high affinity” TCR. The idea that increased downregulation of the high affinity TCR leads to lower cytokine production in response to low amounts of antigen doesn't really make sense to this reviewer. For the revision, we repeated again the dose response and combined this new data with the prior experiments (see revised Figure 6f). As the difference in functional avidity was rather minor in the original submission, and the new data support that there is not a significant difference, we now conclude that the functional avidity of 7431 TRex vs. 1045 TRex T cells is similar despite clear differences in tetramer staining. Since we also observe increased antigen sensitivity in P14 TRex T cells vs. P14 transgenic (see new Figure 8), we now conclude that targeting a TCR to *Trac* improves antigen sensitivity. We have removed the statement that this is related to downregulation of the TCR from the revised manuscript as we agree that this is correlative but not necessarily related. These new data are discussed in the results on pages 10-11 (1045 model) and pages 12-13 (P14 model). We also discuss these findings on pages 17-18 of the discussion.

6. “Unexpectedly, antigen encounter induced P14 TCR downregulation to a greater extent as compared to both 7431 and 1045 T cells (Fig. 5K-L).” This could be explained by a proportion of the TRex cells having endogenous beta, whereas P14 shows allelic exclusion.

As mentioned above, due to comments from Reviewer #3 (see point #2 below), as well as advice from the Editor, our original comparison of 7431 and 1045 TRex T cells to P14 transgenic T cells was not ideal because TCR specificities were different. Therefore, in the revised manuscript, we no longer compare the 1045 and 7431 TRex T cells to P14 transgenic T cells and thus the above statement is removed from the text. We have revised Figure 6 in the new manuscript to remove the P14 transgenic T cell data in all panels, except in Figure 6d where it serves as a negative control for Msln tetramer staining.

7. “Over 90% of CD4 T cells expressed the Msln specific TCR in 1045 and 7431 mice (Fig. 6A). In contrast, only 30-40% of CD4 T cells expressed the gp33-specific TCR in P14 mice (Fig. 6A).” Unfortunately, this is meaningless since they define the Msln-specific TCRs based on Vb9 expression, but the P14 TCR based on Va2 expression (they could have used Vb8). The result for CD4 cells in P14 is due to rearrangement of endogenous Va-regions, impossible in the TRex mice. However, I suspect they would get a result similar to that for the Msln-specific TCRs if they tested for anti-Vb8 in the P14 cells. It is still possible in my view that there are plenty of other Vb elements expressed in the Msln-specific T cells.

As stated above (points #3 and #6), we removed the P14 transgenic comparison to the 1045 and 7431 TRex T cells in the revised manuscript. We include a direct comparison of P14 TRex to P14 transgenic T cells and stain for both Va2 and Vb8 in CD4 T cells in new Figure 8b-c. We show that more CD4 T cells express the P14 TCR in TRex mice vs. transgenic mice, which is consistent with additional endogenous TCR α elements in P14 transgenic T cells. We also address endogenous Vb. See response to point #1 above.

8. “While CD4 T cells isolated from 7431 and 1045 KI mice expressed similar TCR based on Vb9 staining (Fig. 5L), CD4 T cells from 1045 mice stained brighter for Msln tetramer compared to CD4 T cells from 7431 mice (Fig. 5L).” Fig 5L and K are misnamed. Really parts of Fig 6.

Thank you for noticing this error. We have corrected this mistake in the revised manuscript (revised Figure 6l-n).

9. The finding that "...that Tregs accumulate during in vitro expansion of P14 T cells (with RV expression of Msln-specific TCRs) (Fig. 2G-H)." can be explained by the P14 cells having no endogenous b-chains but also no allelic exclusion of endogenous alpha chains, so a lot of possibilities for different pairings.

Since we added a substantial amount of new data to address the major concerns, we selected to move the Treg data to Supplementary Figure 5 in the revised manuscript to adhere to the journals figure limits. Indeed, P14 transgenic T cells, although having no endogenous beta chains, can express additional endogenous V α chains and therefore the Treg expansion may be dependent on CD4 T cells that express a different V α chain and our data supports this (Supplementary Fig. 5g). However, since Tregs expand or differentiate more readily from P14 spleens than wild type (WT) spleens (see Supplementary Figure 5e-f in the revised manuscript), endogenous TCR α expression does not seem to be the only explanation. We have addressed in the results section in the revised manuscript as follows: *"In contrast to T cells from WT mice, T cells from P14 transgenic mice activated with α CD3+ α CD28 and IL-2 exhibited increased frequency of Foxp3+CD25+ Tregs (Supplementary Fig. 5e-f), and many of these did not express V α 2, the P14 TCR α chain (Supplementary Fig. 5g). Thus, endogenous TCR expression may be a prerequisite but not the only factor contributing to disproportionate Treg accumulation in P14 transgenic mice."* (page 9)

10. The results of Fig 6F,G (not 5 as stated) in fact imply that the P14 cells that turn into Tregs express non-Va2 alpha chains, and it is entirely possible that the Msln-specific Tregs can express other Vb than Vb9 – this really should be checked. Also, the staining of P14 with aVb8, not just aVa2, should be tested.

We address this concern in the revised manuscript. We now include the proportion of CD4+Foxp3+ Tregs, CD4+Foxp3- conventional (Tcon) and CD8+ T cells in the periphery that co-express an endogenous V β with the exogenous V β in 1045+/+ TRex, P14+/+ TRex, P14 transgenic and wild type (WT) mice (see new Supplementary Figures 7j-k). We describe these new data in the text as follows, *"20% of peripheral CD8, CD4 conventional (Tcon) and CD4+Foxp3+ cells co-expressed an endogenous and exogenous V β in P14 TRex mice (Fig. 7p, Supplementary Fig. 7h-j). In 1045^{+/+} Msln^{-/-} TRex mice, 30-40% of peripheral CD8, CD4 Tcon and Treg were dual V β + (Supplementary Fig. 7j-k) revealing variability with respect to endogenous V β among independently derived TRex T cell strains."* (pages 12-13)

Minor points:

10. P5. "...as determined Va2 staining" should be "as determined by Va2 staining"

Thank you. This error is no longer in the revision.

11. P9. Needs editing: "CD4+Vb9+ T cells from 1045 TRex mice die upregulated CD69 following Msln406-414:H-2Db recognition..."

We have updated this phrase in the revised manuscript to better encompass the results and to shorten the text overall to adhere to journal guidelines as follows, *"CD4+V β 9+ T cells from 1045 TRex mice upregulated CD69 and CD25 following antigen recognition, yet proliferation was modest and CD44 was not further increased (Fig. 5g-h)." (page 9)*

12. P10. "... (Fig. 5G), which was inconsistent tetramer staining." Should read "inconsistent with tetramer"

Due the extensive revision in the text, this error is longer in the revised manuscript. We now have the summary statement describing current Figure 6G (was 5G in the original submission) as follows, “*Thus, despite decreased tetramer staining and similar TCR cell surface levels consistent with a lower affinity TCR, 7431 effector T cells exhibit a functional avidity comparable to 1045 effector T cells suggesting Trac targeting may enhance antigen responsiveness of lower affinity TCRs.*” (page 10-11)

Reviewer #3 (Remarks to the Author):

The MS by Stromnes et al describes the generation of mice with a quasi monoclonal TCR repertoire through CRISPR mediated knock in of rearranged TCRA and TCRB chains from two different Msln specific TCRs. The authors discuss that this strategy i) represents an accelerated approach to the generation of TCR transgenic mice as compared to 'classical' approaches and ii) is a way to achieve more physiological expression of TCRs (as compared to conventional TCR transgenics) with 'improved fidelity'.

1. Point i) is thoroughly addressed in the first half of the MS and the technology and methods that were applied seem well described, also in comparison to introduction of the very same TCRs into mature T cells through retroviral transduction. This is interesting, and to my knowledge the first description of the heritable and orthotopic introduction of TCRs into the mouse germline using this approach. A major achievement. We appreciate your thoughtful comments regarding our manuscript.

2. Point ii) deals with the question of whether the gene replacement strategy results in T cells with 'more physiological' features (vis-a-vis conventional TCR transgenics) regarding timing of TCR expression, selection into distinct T cell lineages and responsiveness to cognate antigen. Unfortunately, the authors in this context compare features of the newly generated Trex systems with existing TCR transgenic systems of an unrelated TCR specificity (mostly the P14 TCR, and in a few instances the OTI system). However, it is very questionable whether these results justify the statement that TCR exchange results in 'higher fidelity' of the emerging T cells. First, it remains somewhat unclear what 'fidelity' is. Second, and way more importantly, using Trex cells and conventional transgenic cells with different TCRs is not suited to elucidate and compare fundamental physiological features (selection, response to antigen) of cells that have been generated in one or the other way. The only way to systematically do this would be to work with a single, well-defined TCR, for instance to 'recreate' the P14 TCR as Trex system. Otherwise, it will not be possible to make any conclusive statements regarding the question whether differences between conventional TCR transgenic cells and Trex cells may actually relate to the methodology that was used to generate these systems or may instead merely reflect inherent features of two distinct TCRs.

We agree with the reviewer #3 that the comparison of 1045 and 7431 TRex T cells to P14 transgenic T cells was flawed. Therefore, we created a P14 TRex model to directly compare the targeted vs. randomly integrated approach on T cell development and functionality with a fixed/identical TCR. Two new main figures (Figures 7 and 8), two new supplementary figures (Supplementary Figures 7 and 8), and a new Supplementary Table 3 were added to the revised manuscript. These data compare thymocyte maturation (Figure 7) and peripheral T cell function (Figure 8) in the newly generated P14 TRex mice to P14 transgenic and wild type (WT) mice. Thus, the text and discussion of the manuscript have been extensively revised to account for the new data, results, and our interpretations. Due to the new model, our conclusions are more substantiated, and the manuscript is much improved.

We appreciate the discussion of the use of the word fidelity. We use this term for the following reasons:

1. The exogenous TCR is located in the physiological *Trac* locus. Thus, we know where, and how many copies of the TCR is actually expressed.
2. New data rigorously assessing thymocyte development and maturation in 1045 TRex (new Figure 4) and P14 TRex (new Figure 7) show that the TCR is expressed at DN4, similar to WT thymocytes, and different from P14 transgenic thymocytes in which the TCR is expressed at DN1 (new Figure 7). TRex thymocytes undergo all stages of maturation, require MHC I expression for positive selection and 1045 TRex thymocytes are susceptible to negative selection when both copies of *Msln* are expressed (new Figure 4 and new Figure 7).
3. Our studies also support that targeting a TCR to *Trac* increases antigen sensitivity of a low affinity TCR (Figure 6 of revised manuscript) and in the P14 TRex model (Figure 8 of revised manuscript) and sustains T cell function after repetitive antigen stimulations (Figure 2).

3. In sum, the 'technological' part of the manuscript presents exciting data, certainly of substantial interest to the field. I think a careful analysis of T cell selection in the Trex models (perhaps in conjunction with an extended 'biological' focus on a comparison to Trex mice in which *Msln* is concomitantly knocked out) would have been sufficient for an exciting MS. Unfortunately, in its present form, the 'biological' part of the MS (and also the title) presents conclusions that cannot be drawn from comparing unrelated TCR specificities and in my opinion is therefore not suitable for publication. We agree with the reviewer that a key missing aspect in the initial submission was the T cell selection data. During the revision, we had time to increase our breedings and further backcrossing of the 1045 TRex colony to thoroughly assess thymocyte maturation and exogenous TCR expression from *Msln*^{+/+}, *Msln*^{+/-} and *Msln*^{-/-} backgrounds (see new Figure 4, new Supplementary Figure 4). Intriguingly, we show that 1045 T cells undergo negative selection when both alleles of *Msln* are present (Fig. 4b-d). We show that the 1045 TCR can be first detected in DN, specifically DN4, which are likely transitioning to DP (Fig. 4e,f,h) and consistent with timing of initial opening of the *Trac* locus. We also show that MHC class I is required for positive selection of the 1045 TRex TCR expressing CD8 T cells in the thymus (Fig. 4j-l). Additionally, we compare thymocyte maturation/development in P14 TRex to P14 transgenic mice (new Figure 7). Thus, we are hopeful that we have sufficiently addressed these major criticisms. We are grateful to both reviewers as we believe the manuscript is much improved.

Additional point:

4. The MS appears to be written in a very sloppy manner. For instance, already the summary is very difficult to understand. There is an excessive number of typos, and on multiple occasions, numbering of display items in the text and in the figures does not correspond to each other, in particular also in figure legends. Too many to list, and it is not the reviewer's task to proofread a MS. This made reviewing the MS very cumbersome.

We apologize for these errors in the initial manuscript as well as the lack of clarity overall. We substantially revised the text throughout to correct errors and to improve readability. We also had additional authors proof-read the manuscript to help. We hope you find the manuscript to be improved. Thank you for your thoughtful comments.

REVIEWERS' COMMENTS

Reviewer #1 (Remarks to the Author):

The resubmission goes well beyond the initial version of the paper. The comparison of the classical P14 transgenic with the CRISPR-READI P14 makes things much clearer. The thymus analysis is well done.

Their new data showing that endogenous beta chains are expressed (needed to go through beta selection) is good, as is the finding that the endogenous betas may be silenced after expression of the transgenic alpha-beta.

The new finding that tolerance to Msln is gene dosage dependent is interesting and it is good that they are following this up.

Reviewer #3 (Remarks to the Author):

See my previous report.

The MS has been improved substantially through the inclusion of a side-by-side comparison of the P14 TCR when expressed through the novel methodology as compared to conventional TCR transgenesis. I still feel that the authors somehow overstate that TRex T cells may be 'superior' to Tg T cells of the same specificity. To me, the unique 'selling point' of the work is the exciting novel technology to generate heritable alleles encoding a TCR of choice rather than the generation of 'better' T cells. The merit of the work would not be smaller if TRex cells were in fact very similar to Tg T cells rather than 'better' (whatever that may mean...). Along those lines, it should be clearly indicated in Figs 7 and 8 i) whether the P14 Tg strain is +/- (and on Rag+/+ background) and ii) whether the P14 TRex strain is +/+ or +/- (optional: is the presumed higher sensitivity of TRex cells robustly seen with different founders?).

Ludger Klein

Response to Reviews

We are sincerely grateful for the reviewers continued time, thoughtfulness, effort, and critique of our manuscript. We revised the text of the manuscript including the title, abstract, introduction, results, discussion, figure legends, supplementary figures, to address the critiques. We believe the revised manuscript is much improved because of the peer review process and will be impactful for the field. Please see our response below:

Reviewer's comments

Reviewer #1 (Remarks to the Author):

The resubmission goes well beyond the initial version of the paper. The comparison of the classical P14 transgenic with the CRISPR-READI P14 makes things much clearer. The thymus analysis is well done. Their new data showing that endogenous beta chains are expressed (needed to go through beta selection) is good, as is the finding that the endogenous betas may be silenced after expression of the transgenic alpha-beta. The new finding that tolerance to Msln is gene dosage dependent is interesting and it is good that they are following this up.

Thank you for the positive response to our manuscript and helpful critiques.

Reviewer #3 (Remarks to the Author):

See my previous report.

The MS has been improved substantially through the inclusion of a side-by-side comparison of the P14 TCR when expressed through the novel methodology as compared to conventional TCR transgenesis. I still feel that the authors somehow overstate that TRex T cells may be 'superior' to Tg T cells of the same specificity. To me, the unique 'selling point' of the work is the exciting novel technology to generate heritable alleles encoding a TCR of choice rather than the generation of 'better' T cells. The merit of the work would not be smaller if TRex cells were in fact very similar to Tg T cells rather than 'better' (whatever that may mean...).

Ludger Klein

Thank you for your helpful critique and perspective. We revised our manuscript in manner to refrain from any appearance of overstating that TRex T cells may be superior to Tg T cells as follows:

1. We changed the title of the revised manuscript to, "***Germline T cell receptor exchange results in physiological T cell development and function***". In this way, we removed the term 'higher fidelity' as not to suggest that the TRex method is superior to historical methodologies.
2. We edited the final paragraph in the introduction of the revised manuscript to now read as follows, "*In sum, we identify a robust and highly efficient method to generate mice with physiological expression of a desired antigen-specific TCR, allowing for a standardized source of antigen-specific T cells. Further, our characterization of TRex mice reveals novel insights into T cell development and functionality.*" As such, we removed the statement from the final paragraph in the introduction that was in the previous versions, "...that circumvents the shortcomings of historical TCR transgenics".
3. Since historical TCR Tgs are the benchmark for comparison of advantages/disadvantages of the TRex mice, we have retained text that does compare the two approaches in the discussion. However, we replaced the word "limitations" to "considerations" in the discussion as follows, "...**Considerations** of this approach include multiple and random TCR integration into the genome resulting in non-physiologic TCR regulation¹⁻⁵ and premature TCR α and TCR β expression at the DN1 stage impacting thymocyte development⁴⁶."
4. We changed the abstract to replace the word "superior" to "increased", as follows, "...we demonstrate increased avidity of Trac-targeted TCRs over transgenic TCRs, while preserving physiologic T cell development."
5. We changed the last sentence in the discussion as follows, "*Thus, the TRex methodology is an exciting and robust technology to generate heritable alleles encoding a desired TCR and permit investigation into physiological TCR regulation on T cell behavior.*"

Along those lines, it should be clearly indicated in Figs 7 and 8 i) whether the P14 Tg strain is +/- (and on Rag+/+ background) and ii) whether the P14 TRex strain is +/- or +/- (optional: is the presumed higher sensitivity of TRex cells robustly seen with different founders?).

Thank you for these important points. In Figures 7 and 8, both the P14 TRex and P14 TCR Tg mice used in the experiments were on the Rag+/+ background and the P14 TRex mice used for the analysis mice were homozygous for the TCR. The P14 TRex mice represent 3 independent founder breedings to generate the F1 homozygous pups for our study. Although word limit constraints did not permit us to add this specific text to the figure legends, we included a statement in the methods in the Animal section in the revision as follows: "*P14 TRex and P14 Tg mice were on the Rag^{+/+} background. P14 TRex mice were homozygous for the P14 TCR. Three independent P14 TRex founders were bred to generate homozygous F1 pups for the P14 TRex analysis.*" We also add a statement in the discussion as follows, "*Since pups from multiple independent P14 TRex founders were analyzed, the heightened antigen sensitivity appears reproducible.*"